# Shotgun transcriptome, spatial omics, and isothermal profiling of SARS-CoV-2 infection reveals unique host responses, viral diversification, and drug interactions

In less than nine months, the Severe Acute Respiratory Syndrome Coronavirus 2 (SARS-CoV-2) killed over a million people, including >25,000 in New York City (NYC) alone. The COVID-19 pandemic caused by SARS-CoV-2 highlights clinical needs to detect infection, track strain evolution, and identify biomarkers of disease course. To address these challenges, we designed a fast (30-minute) colorimetric test (LAMP) for SARS-CoV-2 infection from naso/oropharyngeal swabs and a large-scale shotgun metatranscriptomics platform (total-RNA-seq) for host, viral, and microbial profiling. We applied these methods to clinical specimens gathered from 669 patients in New York City during the first two months of the outbreak, yielding a broad molecular portrait of the emerging COVID-19 disease. We find significant enrichment of a NYC-distinctive clade of the virus (20C), as well as host responses in interferon, ACE, hematological, and olfaction pathways. In addition, we use 50,821 patient records to find that renin–angiotensin–aldosterone system inhibitors have a protective effect for severe COVID-19 outcomes, unlike similar drugs. Finally, spatial transcriptomic data from COVID-19 patient autopsy tissues reveal distinct *ACE2* expression loci, with macrophage and neutrophil infiltration in the lungs. These findings can inform public health and may help develop and drive SARS-CoV-2 diagnostic, prevention, and treatment strategies.

In March 2020, the World Health Organization declared a pandemic of the coronavirus disease 2019 (COVID-19), an infection caused by severe acute respiratory syndrome coronavirus 2 (SARS-CoV-2)[1]. Since the start of the pandemic, more than >100 million cases and 2 million deaths have been reported (https://coronavirus.jhu.edu), with an especially high burden of early cases in New York City (NYC). Genomic epidemiology efforts have already played a crucial public health role in confirming community spread of SARS-CoV-2 in the USA[2–4] as well as in China[5] and around the world[6–8]. However, standard approaches to viral profiling (i.e., qRT-PCR or targeted methods) fail to provide information on either the host immune response or microbial co-infections, both of which might impact clinical presentation of COVID-19 and provide directions for therapeutic intervention and management.

Studies of SARS infection have shown that regulation of the innate immune response is associated with the development of adaptive immunity and disease severity. Moreover, a robust inflammatory response is characterized by an upregulation of cytokines including IL-6, IL-10, and MCP-1 in tissues and serum, as well as infiltration of infected tissues by inflammatory cells such as macrophages[9]. SARS-CoV-2 infection studies have shown that viral load peaks during the first week of illness, which may account for the high transmissibility of SARS-CoV-2[2]. Furthermore, antibody profiling by[10] demonstrates both IgM and IgG antibodies began to increase around the 10th day after symptom onset, and most patients had seroconversion within the first 3 weeks of infection, underscoring the need for rapid testing in the acute phase of infection. Moreover, phenotyping of COVID-19 patients has also been shown to identify molecular signatures to distinguish severity of symptoms[11,12], but limited information about the association of the viral sequences and outcome exists, nor viral propagation within human tissues.

To better understand the impact and progression of SARS-CoV-2 infection, we applied a multi-platform and molecular diagnostic approach to samples collected during the outbreak in NYC. Out of concern for the scalability and sensitivity of standard diagnostic assays (qRT-PCR), we developed and validated a rapid reverse transcription loop-mediated isothermal amplification (RT-LAMP) assay to detect SARS-CoV-2 infection from nasopharyngeal swab specimens and oropharyngeal swab lysates. We simultaneously developed a large-scale host and viral profiling platform employing shotgun metatranscriptomics (total RNA-seq). We applied this total RNA-seq platform to 732 clinical samples, including 669 confirmed or suspected COVID-19 cases at New York-Presbyterian Hospital-Weill Cornell Medical Center (NYPH-WCMC). In addition, we report an observational study of renin–angiotensin–aldosterone system inhibitors and severe COVID-19 outcomes using data ($n = 50,821$) from NYPH-WCMC and New York-Presbyterian Hospital-Columbia University Irving Medical Center (NYPH-CUIMC).

Our results define and map the viral genetic features of the NYC outbreak and associate specific host responses and gene expression perturbations with SARS-CoV-2 infection. We also link findings from host transcriptomics to findings from clinical data related to the angiotensin converting enzyme (ACE) pathway using patients at two NYPH sites. Finally, we used spatial transcriptomics to map ACE expression in SARS-CoV-2 infected lungs from COVID-19 patient autopsies. We have made these data available in the database of genotypes and phenotypes (dbGAP) and also in an interactive analytics portal to enable others to explore additional genomic, transcriptomic, and clinical covariate associations (https://covidgenes.weill.cornell.edu/).

## Results

**Validation of LAMP for detection of SARS-CoV-2.** We developed a colorimetric assay to quickly detect and quantify SARS-CoV-2 viral load in patient samples, using a set of LAMP primers and simple single-tube protocol (Fig. 1a, b). To validate the assay, we first evaluated two synthetic RNAs (see "Methods") whose sequences matched the viral sequences of patients from Wuhan, China and Melbourne, Australia (Supplementary Fig. 1). The reaction output was measured at 0-, 20-, and 30 min intervals (Fig. 1c) before the samples were heated to 95 °C for inactivation. LAMP fluorescence correlated closely with SARS-CoV-2 RNA viral copies (Fig. 1d), with an overlap of the median signal from negative controls at lower levels (0–10 total copies per reaction) of viral RNA ($n = 10$). The LoD was found to be between 5 and 25 viral total copies for the dual primer, single-tube reaction (N gene and E gene), and this was then replicated to show a similar LoD on the second synthetic control (Supplementary Fig. 1b).

We next used a set of 201 samples from known or suspected COVID-19 cases that were tested for SARS-CoV-2 by a commercial qRT-PCR assay (Altona) that has been a standard clinical test at NYPH-WCMC since early March. These comprised 69 nasopharyngeal (NP) swab samples that tested positive (qRT-PCR positives, Ct < 40) and 132 samples that tested negative (qRT-PCR negative, Ct ≥ 40) (see "Methods"). qRT-PCR positive samples showed a much higher LAMP fluorescence than qRT-PCR negative samples. Analysis of Receiver Operator Characteristic (ROC) curves yielded an overall sensitivity of 95.6% and specificity of 99.2% (Fig. 1e, Supplementary Fig. 2). These results were also confirmed with a capture-based SARS-CoV-2 panel from Twist (Supplementary Fig. 2b, c). Of note, we obtained similar performance on bulk oropharyngeal swab lysate (Supplementary Fig. 3), including increasing reaction sensitivity as a function of viral copy number, but with deteriorating performance Ct > 30. We observed higher LAMP sensitivity at higher viral loads, as determined by qRT-PCR Ct values (Supplementary Fig. 4).

**Shotgun metatranscriptomics platform for viral and bacterial detection.** To further investigate the biological characteristics of qRT-PCR positive and negative specimens, as well as to compare to RT-LAMP, we developed a shotgun metatranscriptomics platform utilizing total RNA-seq (RNA-sequencing with ribosomal RNA depletion) to profile all RNAs from patient specimens (Fig. 1a). We sequenced 857 RNA-seq libraries (Fig. 2) across 732 specimens from 669 patients treated for influenza-like illness (ILI) at NYPH-WCMC to an average of 63.2M read pairs per sample. This included 215 qRT-PCR positive samples (201 of which were also tested with LAMP), 17 positive (Vero E6 cells), and 33 negative (buffer) controls. To assess the ability of deep shotgun transcriptomics to identify qRT-PCR false negatives, including those that might arise from uncharacterized SARS-Cov-2 variants, we sequenced 517 qRT-PCR negative samples. Among these qRT-PCR negative samples, 311 were tested for the standard clinical respiratory virus panel (BioFire, which included several common cold coronavirus strains and influenza virus).

First, the taxonomic composition of samples was detailed by aligning total RNA-seq reads to the human reference genome GRCh38 and NCBI reference databases (see "Methods"). Kraken2 classification of non-human sequences revealed abundant bacterial and SARS-CoV-2 RNA among qRT-PCR positive samples and infrequent mappings to fungi, archaea, or other viruses (Fig. 2a). Positive controls and clinical samples with medium/high-viral load-consistent qRT-PCR Ct values were significantly ($p < 1 \times 10^{-16}$) enriched in SARS-CoV-2 genome alignments (median reads per kilobase per million mapped reads, RPKM) relative to

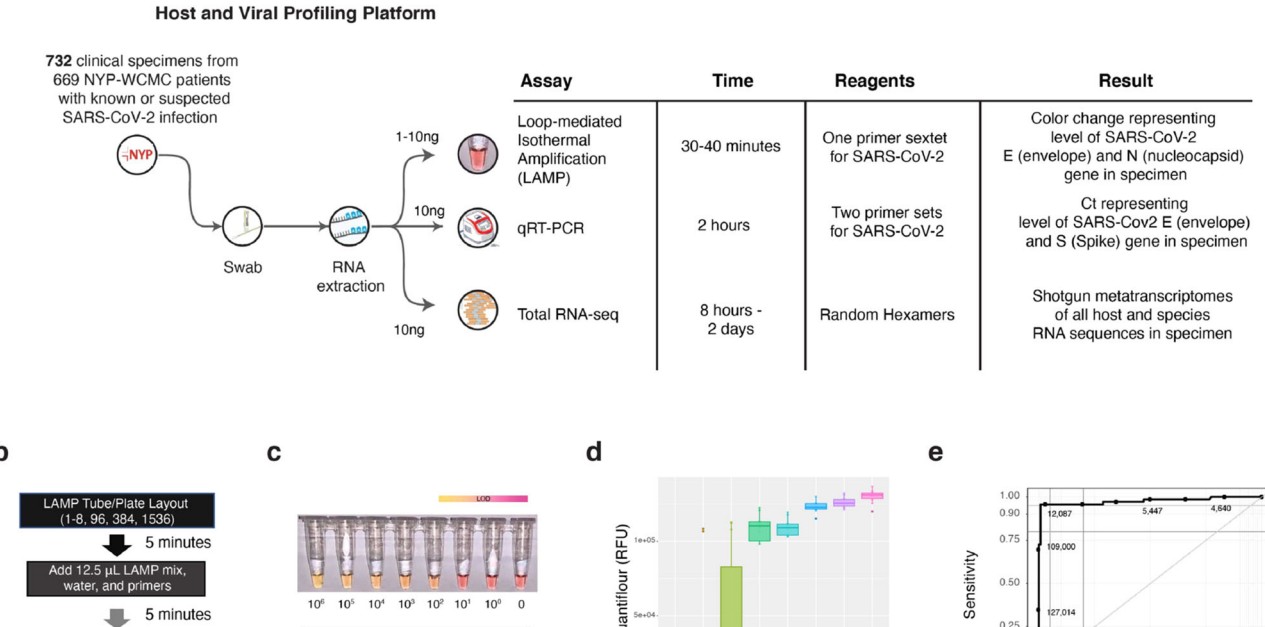

**Fig. 1 Sample processing, the loop-mediated isothermal (LAMP) reaction and synthetic RNA validation. a** Clinical samples collected with nasopharyngeal (NP) swabs were tested with RNA-sequencing, qRT-PCR, and LAMP. **b** The test samples were prepared using an optimized LAMP protocol from NEB, with a reaction time of 30 min. **c** Reaction progress was measured for the Twist SARS-CoV-2 synthetic RNA (*MT007544.1*) from 1 million molecules of virus[4], then titrated down by $\log_{10}$ dilutions. The colorimetric findings of the LAMP assay are based on a yellow to pink gradient with higher copies of SARS-CoV-2 RNA corresponding to a yellow color. The limit of detection (LoD) range is shown with a gradient after 30 min between 10 and 100 viral copies (lower right). **d** Replicates of the titrated viral copies using LAMP, as measured by QuantiFluor fluorescence over 201 patient samples. **e** The sensitivity and specificity of the LAMP assay from 201 patients (132 negative and 69 positive for SARS-CoV-2, as measured by qRT-PCR). Thresholds are DNA quantified by the QuantiFluor.

negative controls and qRT-PCR negative clinical samples (Fig. 2a). LAMP fluorescence, qRT-PCR Ct values, and total-RNA Seq RPKM showed consistent estimates of SARS-CoV-2 viral abundance across these three technologies and 201 clinical specimens ($R_{seq\_vs.\_Ct} = -0.84$, $R_{seq\_vs.\_lamp} = 0.82$, $R_{lamp\_vs.\_Ct} = -0.80$) (Supplementary Fig. 4a, b). Analysis of SARS-Cov-2 read coverage across 517 qRT-PCR negative samples revealed 7 (1.3%) samples with more than 0.01% of reads matching SARS-CoV-2. In summary, these results indicate close concordance with SARS-Cov-2 viral load across diverse diagnostic platforms, with rare examples of occult SARS-Cov-2 infection among qRT-PCR negative cases.

Next, an analysis of total RNA-seq sequences showed the presence of commensal species and viruses across both qRT-PCR positives and negative samples (Fig. 2a). We found a statistically significant higher load of other (i.e., non-SARS-CoV-2) respiratory viruses in qRT-PCR negative vs. positive samples (Wilcoxon test, $P = 0.0008$) (Fig. 2b). Oral and airway commensal taxa were compared between the high-viral load group and other patient categories using linear modeling in log-normal space using MaAsLin2 package (see methods) and relative to the negative controls. This includes a correction for the SARS-CoV-2 or *Homo sapiens* matching reads, which can show variable depth of coverage due to higher SARS-CoV-2 fraction in the high-viral load group. After this correction, 17 species were significantly depleted in COVID-19 patients, including *A. xylosoxidans*, *E. faecalis*, and *L. bacterium* (Supplementary Fig. 5).

Among respiratory viruses discovered across all patients (COVID+ and COVID−), we found frequent Influenza A

(23% of viral positive cases), rhinovirus A (16%), and human metapneumovirus (12%). Overall, we found close concordance between these results and the findings of a transcriptomics based viral detection to the results of a standard (BioFire) respiratory pathogen PCR panel performed within 7 days of the NP swab used for RNA sequencing ($N = 356$ patients across 404 tests) (Fig. 2c). The metatranscriptomics platform yielded an Area Under the Receiver Operator Characteristic (AUROC) curve of 0.890 (F-score = 0.725), and an accuracy of 0.994, sensitivity of 0.758, specificity of 0.997, and precision of 0.694 (see methods). These data also enabled an examination of co-infection within COVID+ patients. Among qRT-PCR positive cases, only 7 (3.2%) harbored sequences mappable to other respiratory viruses. Among these, we found human coronaviruses 229E, NL63 and HKU1, influenza A, and human mastadenovirus and metapneumovirus (Fig. 2b). In summary, these results indicate that common respiratory viruses were frequently implicated in many Influenza-like Illness (ILI) cases that were COVID-, and demonstrate low incidence of co-infection of SARS-CoV-2 with other respiratory viruses (3.2%), matching results (3%) from other studies[13].

**SARS-CoV-2 assemblies from shotgun metatranscriptomes.** The abundance of SARS-CoV-2 alignments from total RNA-seq was sufficient to provide >10× coverage of the viral genomes and yield high quality, full-length assemblies for 155 samples (Supplementary Fig. 6). When examining the viral genomes, we identified a total of 1147 instances of 165 unique variants across

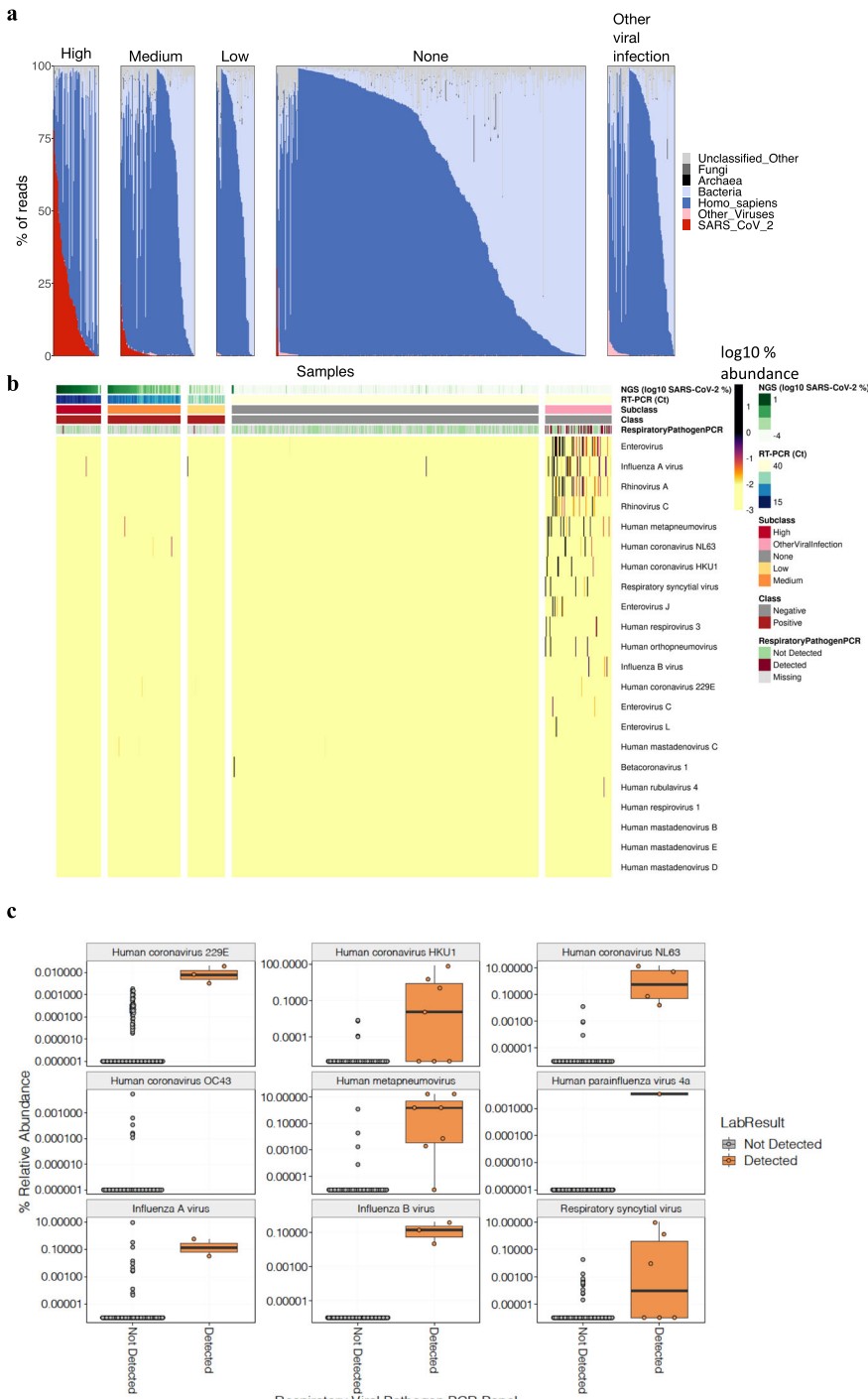

**Fig. 2 Full transcriptome profiles of SARS-CoV-2 Patients with NGS, qRT-PCR, and LAMP. a** Reads mapping to SARS-CoV-2 (red), other viruses (pink), human (blue), bacteria (light blue), archaea (black), fungi (dark gray), across samples with high/medium/low viral load according to qRT-PCR ("High", "Medium", and "Low", respectively), qRT-PCR negative samples with no detection of other respiratory viruses ("None"), and qRT-PCR negative samples in which other respiratory viruses were detected by RNA sequencing and/or by a BioFire panel ("Other viral infection"). **b** Heatmap of abundance of a selection of viral pathogens across samples. **c** BioFire viral validation of the detection of viral pathogens using RNA sequencing across 669 samples. Box plots to compare relative abundance of viral pathogens in RNA sequencing between samples that were found as positive or negative for each virus in a BioFire PCR panel. Box plots show the median as center, first and third quartiles as the box hinges, and whiskers extend to the smallest and largest value no further than the 1.5× interquartile range (IQR) away from the hinges.

155 assemblies, including 1143 single nucleotide variants (SNVs) and four deletions. We compared the 155 full-length NYPH-WCMC sequences with 46,581 SARS-CoV-2 sequences obtained from a recent GISAID build (downloaded on 6/16/2020) using the Nextstrain pipeline[7] (Fig. 3a, see Methods). We generated

full-length SARS-CoV-2 assemblies for 9 of 517 (1.7%) of samples that were found negative by qRT-PCR (Supplementary Fig. 6a). Each of these demonstrated a high RPKM viral load, an abundance of reads evenly covering the SARS-CoV-2 genome, and high (>0.5) variant allele fractions (VAFs) of SNV's commonly

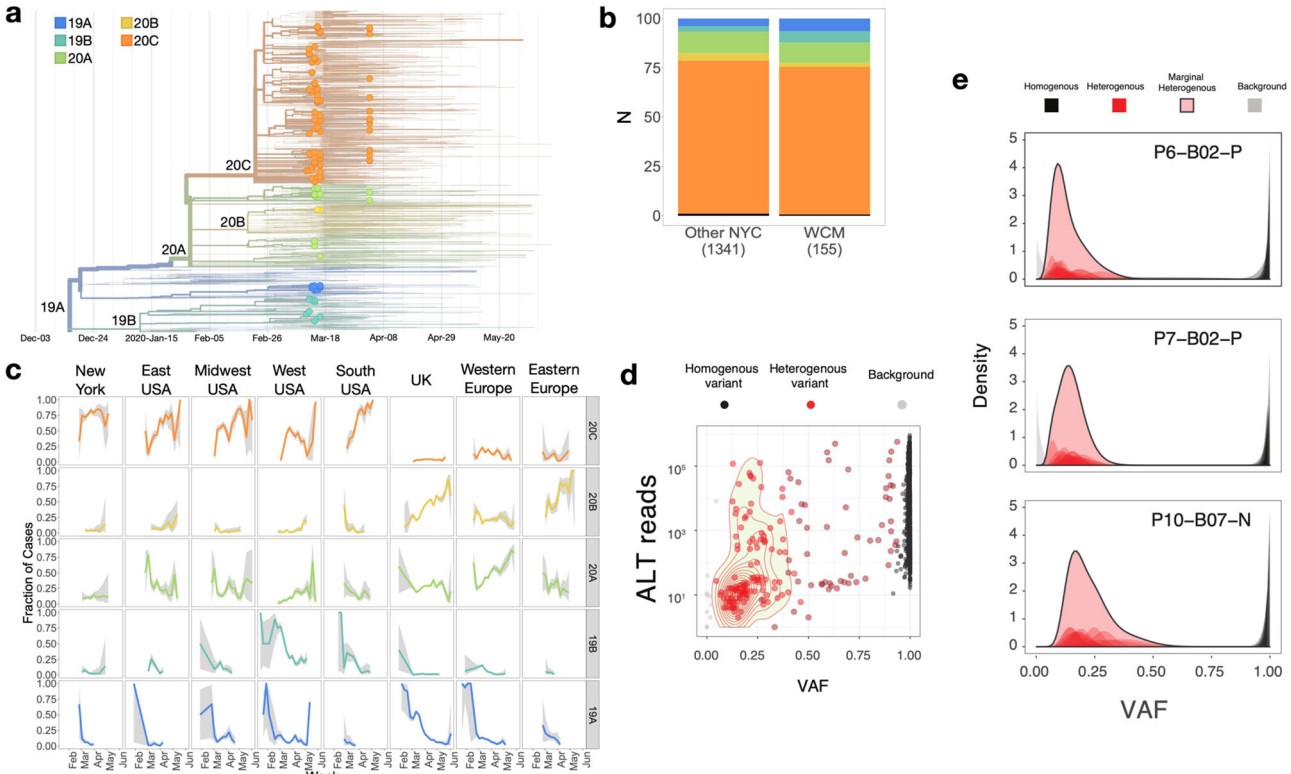

**Fig. 3 Viral genome assemblies and variants. a** Time-resolved phylogenetic tree of WCM samples and GISAID samples. Nodes corresponding to the 155 WCM strains are marked with circles. Branches and nodes are colored according to the Nextstrain clade affiliation. **b** Clade distribution amongst WCM strains and other NY strains from GISAID. **c** Longitudinal distribution of clade assignments. Data points represent the portion of the sequences on GISAID matching to each clade in a given week in each of the 8 regions. Lines show the mean and gray areas around the curve represent 95% confidence intervals. **d** Variant allele frequencies (VAF, x-axis) for alternative alleles (y-axis) were calculated for all variants across viral strains, with heterogenous (het, 5% < × < 95%, red) variants shown as well as homogenous (black) variants with VAF > 0.95. **e** The distribution and density of the VAF for three exemplar samples are shown relative to their variant type (top).

seen in other positive samples and GISAID (Supplementary Fig. 6a). These results indicate that qRT-PCR false negatives can occur, but are rare.

We then used a New York State-focused build of Nextstrain resulting in the incorporation of our 155 genomes with 3541 sequences from GISAID (Fig. 3a). Analyzing worldwide groupings annotated by the Nextstrain database, we found a high proportion (>74%, 116/155) of our assemblies were associated with clade 20C (Fig. 3b), which overall represents 17% of strains in the GISAID build. Our metatranscriptomic assembly-based clade assignments for NYC strains were consistent with those obtained by parallel targeted-sequencing based studies from the same period (Fig. 3b). Investigating the geographical and temporal distribution of non-NYC samples in 20C, we found a global distribution with a likely origin in Western Europe (Fig. 3c). The first noted cases of 20C in GISAID were recovered in France on 2/21/2020, but since then, this clade has represented a minority (<20%) of Western European cases. In contrast, 20C represented a steady majority of (>70%) of NYC cases from March to May 2020. Interestingly, 20C was relatively rare (<50%) in other regions of the US in March, but increased through June. The most striking increase was in the southern USA, where 20C began under 30% prevalence in March and rose to above 80% prevalence in May and June. The SNVs defining the 20C clade were non-synonymous variants targeting genes encoding the nonstructural protein 2 (NSP2), the viral replicase, and ORF3a, a poorly characterized SARS-CoV-2 protein with putative roles in inflammation[14]. These observations refine the results of a previous study that associated NYC cases with clade A2a[3]. This

clade is defined by the presence of the Spike p.D614G mutations and comprises a superset of 20C.

While the majority of the known variation in SARS-Cov-2 represents SNVs, one hundred distinct deletion variants have been reported in previous analyses[15]. Of the four deletions we detected in our assemblies, three represented the same in-frame deletion in nsp1 (p.141_143del) (Supplementary Fig. 7a). This deletion was well supported by reads in our samples (Supplementary Fig. 7b). While it was not exclusive to genomes from our study, we only detected this deletion in 168 GISAID genomes (0.3%). Interestingly, rather than forming a single monophyletic subclade, genomes carrying this deletion spanned all major SARS-Cov-2 clades (19A: 12, 19B: 4, 20A: 68, 20B: 30, 20c: 53, none: 4) suggesting that this deletion may have evolved many times during SARS-CoV-2 community spread (Supplementary Fig. 7c). In addition, while the dominant (171/172) version of this deletion involved the in-frame deletion of nucleotide positions 686-694, a single genome included an alternative in-frame deletion of positions 684–692, resulting in the same amino-acid deletion.

Variant calling of assemblies and RNA-seq reference alignments showed most (87.5%) variants have VAFs >0.95 and high (>100) numbers of variant-supporting reads. Analysis of VAF posterior probability distributions identified a subset of alleles whose VAFs were confidently (probability > 0.95) above 0.05 but below 0.95 (Fig. 3d, e). We labeled these alleles as "het" (heterogenous) variants and refer to the remaining (high VAF) consensus assembly variants as "homogeneous." Many of these het variants were associated with robust read support despite

frequently having VAFs below 0.05 and not being incorporated into the consensus assembly sequence. (Supplementary Fig. 6b). These results indicate that SARS-CoV-2 samples frequently harbor minority viral populations, which can be detected and characterized with shotgun metatranscriptomics. The relatively large numbers of these het variants (>10) in a subset of cases, presumably arising during the course of a single infection and then rising to a reasonably high oligoclonal VAFs (0.1–0.2) level, suggests that this intrahost diversification may be rapid and also associated with positive selection.

**Defining the SARS-CoV-2 host transcriptome**. We then leveraged the total RNA-seq profiles to better understand the host transcriptome during SARS-CoV-2 infection. Cell proportion analyses using the MUSIC deconvolution algorithm showed enriched proportions of cell types spanning goblet, ciliated airway, and epithelial cells across all samples (Supplementary Fig. 8a, b). Differentially expressed genes (DEGs) associated with SARS-CoV-2 infection were calculated using limma voom and DESeq2 (see "Methods"). Overall, 757 significant DEGs ($q < 0.01$, >1.5-fold change) were found in the qRT-PCR positive vs. negative samples (Supplementary Data 3), spanning 350 upregulated DEGs and 407 downregulated DEGs, and a total of 8851 unique genes ($q < 0.01$, >1.5-fold change) (Fig. 4).

Differentially expressed host genes indicated a wide range of antiviral responses, including a common interferon response across all ranges of viral levels, which was significantly higher when compared to SARS-CoV-2 negative samples that harbored other respiratory viruses (Fig. 4a, b). Notably, host cells showed an increase in angiotensin converting enzyme 2 (ACE2) expression (Fig. 4b) ($p$ value $= 1.4 \times 10^{-9}$), which is the SARS-CoV-2 cellular receptor[16]. This critical gene for viral entry[17] exhibited an expression level concomitant with the higher levels of SARS-CoV-2 virus, along with IFI27 (Interferon Alpha Inducible Protein 27, $p < 2.2 \times 10^{-16}$) and IFI6 (Interferon Alpha Inducible Protein 6, $p < 2.2 \times 10^{-16}$). The DEGs also included HERC6 (HECT Containing E3 Ubiquitin Protein Ligase Family Member 6), which aids Class I MHC-mediated antigen processing and Interferon-Stimulated Genes (ISGs) (Fig. 4c), underscoring the impact of the virus on these cells' immune response[18]. Also, a subset of cytokines (CXCL10, CXCL11, and CCL8) showed the highest spike of expression in the higher viral load sub-group, matching previous results from animal models and infected cells[19]. Of note, these expression patterns significantly overlapped with those taken from an independent cohort (UCSF) of 286 NP swabs also processed by RNA-seq (see Methods and Supplementary Fig. 8c, $p = <2.2 \times 10^{-16}$). The host transcriptome that exhibited the greatest amount of DEGs were those with the highest viral titer (Supplementary Fig. 9).

Downregulated genes and those with a negative enrichment score (NES) were functionally distinct (Fig. 4d). This included a significant decrease in gene expression for the olfactory receptor pathway genes ($q$ value $= 0.0005$, Supplementary Data 4), which is consistent with a COVID-19 phenotype wherein patients lose their sense of smell. Other downregulated genes included the transmembrane serine protease TMPRSS-11B, which regulates lung cell growth[20] and ALAS2, a gene which makes erythroid ALA-synthase[21] that is found in developing erythroblasts (red blood cells). ALA-synthase plays an important role in the production of heme TRIM2 E3 ubiquitin ligase induced during late erythropoiesis, which indicated a connection to hematological and iron (heme) regulation during infection. Accordingly, genes in a related biological network were significantly enriched based on Gene Ontology (GO) pathways for iron regulation ($q$ value $= 0.04$, Supplementary Data 4). Both the upregulated and

downregulated gene expression differences were distinct from those of house-keeping genes, which stayed mostly stable during infection (Supplementary Fig. 10).

**ACE inhibitor/angiotensin receptor blocker usage correlates with COVID-19**. Given our observation of increased ACE2 gene expression in patients with high SARS-CoV-2 viral load, we investigated the interplay of receiving pharmacologic angiotensin converting enzyme inhibition (ACEI) or angiotensin receptor blockers (ARBs) for hypertension and clinical features of COVID-19. Since ACE2 expression can be increased in patients taking ACEIs and ARBs[22], the observed correlation of viral titer with ACE2 expression may be attributed to the pre-infection use of such inhibitors, which is common in older patients and those with comorbidities[23].

To address this, we analyzed ACEI/ARB use and severe COVID-19 outcomes in an observational cohort of individuals ($n = 50,821$) suspected of SARS-CoV-2 infection at New York-Presbyterian Hospitals, comprising 23,170 patients from Columbia University Irving Medical Center (NYP-CUIMC) and 27,651 patients from Weill Cornell Medical Center (NYP-WCMC; Table 1). At both sites, we found evidence that ACEI/ARB use is associated with lower rates of intubation (CUIMC Hazard ratio, HR = 0.79, 95% confidence interval, CI: 0.68–0.91, WCMC HR = 0.62, CI: 0.48–0.81) and lower risk of death (CUIMC HR = 0.66, CI: 0.58–0.75, WCMC HR = 0.73, CI: 0.56–0.95) following confirmed SARS-CoV-2 infection (Fig. 5). Each comparison used propensity matching (PSM) and Cox proportional hazards models with covariate adjustments for age, sex, race, ethnicity, drug indications, and relevant comorbidities (see "Methods"; Table 1).

ACEI/ARB have a number of indications in addition to hypertension, so confounding variables remains a challenge. To address this issue, we repeated the analysis in a cohort of individuals with recent exposure to one of four antihypertensive medication classes: ACEI/ARB, beta blockers (BB), calcium channel blockers (CCB), and thiazide/thiazide-like diuretics (THZ)). Evaluating each drug class separately using PSM and covariate adjustment, we found the rate of intubation was consistently lower for ACEI/ARB exposure (CUIMC HR = 0.74, CI: 0.64–0.85, WCMC HR = 0.73, CI: 0.56–0.95) and higher for THZ exposure (CUIMC HR = 1.96, CI: 1.67–2.31, WCMC HR = 1.90, 1.42–2.55). Conversely, we found no statistically significant associations for CCB (CUIMC HR = 0.92, 0.81–1.05, WCMC HR = 1.04, CI: 0.83–1.30), and only weak evidence of positive associations between intubation and BB exposure (CUIMC HR = 1.16, CI: 1.03–1.31, WCMC HR = 1.04, CI: 0.84–1.27). Meanwhile, we found consistently lower rates of death among patients with ACEI/ARB (CUIMC HR = 0.82, CI: 0.71–0.93, WCMC HR = 0.67, CI: 0.52–0.87), BB (CUIMC HR = 0.81, CI: 0.71–0.91, WCMC HR = 0.74, CI: 0.61–0.91), and CCB (CUIMC HR = 0.63, CI: 0.55–0.72, WCMC HR = 0.69, CI: 0.55–0.86) exposures, while THZ exposure had harmful associations (CUIMC HR = 1.03, CI: 0.87–1.22, WCMC HR = 1.43, CI: 1.04–1.96). In summary, among SARS-CoV-2-infected individuals with recent exposure to antihypertensive drugs, ACEI/ARBs were the only class which were associated with lower risk of intubation, while ACEI/ARB, BB, and CCB use was negatively associated with death. (Complete estimates are provided in Supplementary Data 6).

**Spatial expression data**. We next used a spatial transcriptomic technology (GeoMx) to characterize the distribution of a set of >1800 genes (the GeoMx Cancer Transcriptome Atlas probe set, or CTA) across tissues from four COVID-19 patients who died

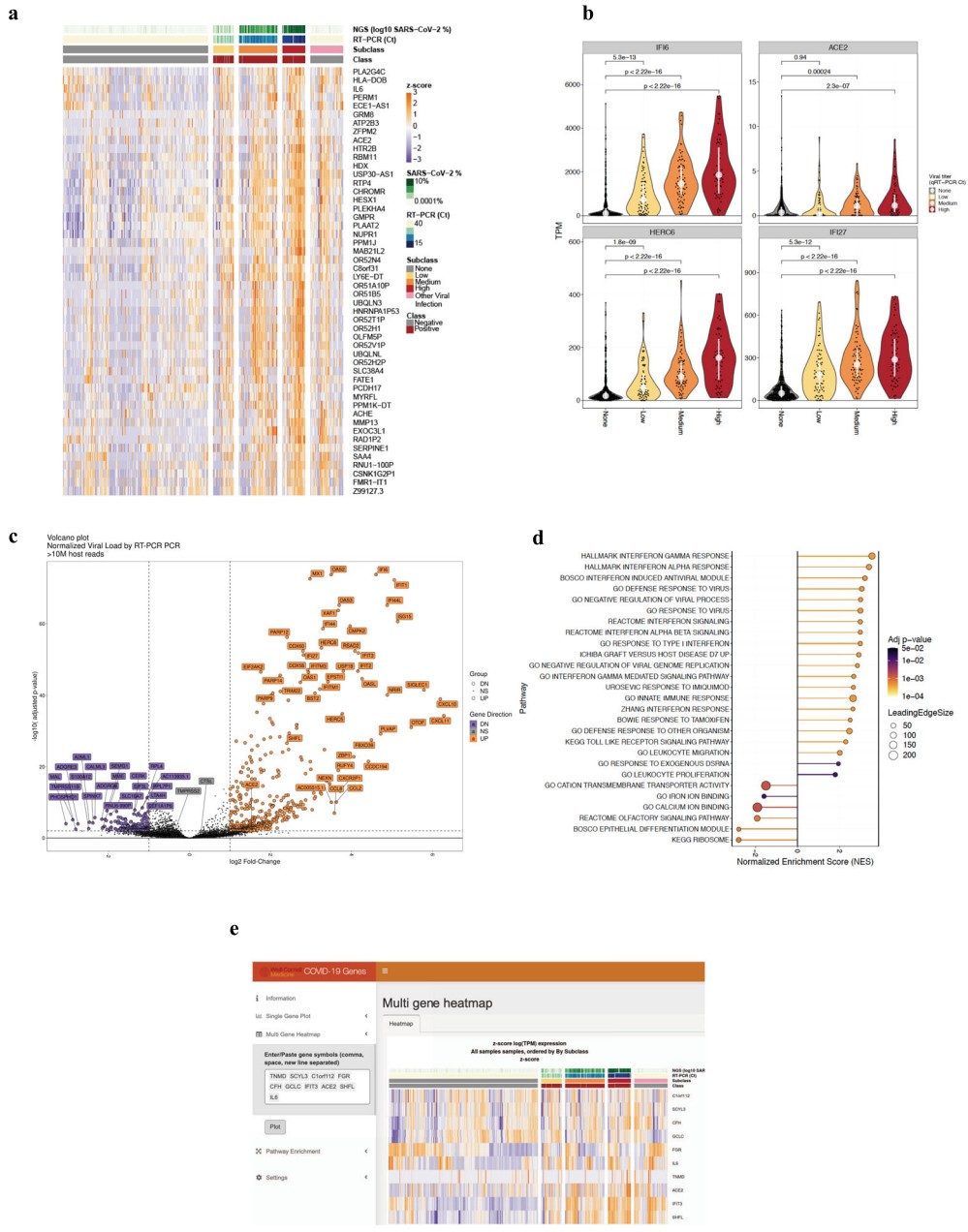

**Fig. 4 Host transcriptome responses to SARS-CoV-2. a** Samples were quantified by RNA-seq (log₁₀ SARS-CoV-2% of reads), and qRT-PCR (Ct values) to create a three-tier range of viral load for the positive samples (class, red) and those samples with other viral infections that were SARS-CoV-2 negative (gray). Differentially expressed genes showed upregulated (orange) genes as well as downregulated (purple) genes compared to non-viral samples. **b** Upregulated genes, with violin plots for all samples, include *IFI6*, *ACE2*, *SHFL*, *HERC6*, *IFI27*, and *IFIT1*, based on data from (**c**), which is the total set of DEGs, shown here as a volcano plot, with a core set of upregulated genes (orange) distinct from the set of downregulated genes (purple), compared to genes that are not significantly differently expressed (gray) in any comparison (Statistical tests by negative binomial model in DESeq2, multiple testing adjusted *p* value < 0.01, |logFC| >0.58). **d** GSEA enrichment of significant pathways, with color indicating statistical significance and circle size the number of genes on the leading edge. **e** Screenshot of the WCM COVID-19 Genes Portal, an interactive repository for mining the human gene expression changes in the data from this study (https://covidgenes.weill.cornell.edu).

and then received an autopsy (Fig. 6); these were compared to three normal lung donors (excess lung material from healthy lung transplant donors). Samples were processed using the Nanostring GeoMx protocols (see methods), including illuminating and UV-cleaving the antibody-linked probes for Areas of Interest (AOIs) within the patients' lung FFPE sections and alveoli, followed by Next-Generation Sequencing (NGS) profiling on the Illumina NextSeq500. These data showed spatially-restricted expression of ACE2 at the perimeter of the alveoli (Fig. 6b), as well as the presence of the SARS-CoV-2 virus in some of the same sections.

We then used cellular deconvolution methods based on the CTA data across the 107 total AOIs (69 COVID+ and 38 COVID−) to characterize the types of cells present in the lung. The cell proportions showed an increase in macrophages present in the lung tissue (Fig. 6c) compared to normal donors, particularly for patient #3, where 30–52% of the cells found within the AOIs were macrophages, as well as patient #4 (up to 18%). In contrast, normal lung samples showed <1% across all AOIs tested (*n* = 38 total AOIs). Of note, patient #1 showed up to 3.8% presence of neutrophils in the AOIs profiled, compared to

**Table 1 ACEI/ARB characteristics.**

| Characteristic | CUIMC | | WCMC | |
|---|---|---|---|---|
| | No ACEi/ARB | ACEi/ARB | No ACEi/ARB | ACEi/ARB |
| N (total = 50,821) | 19,299 | 3871 | 25,017 | 2634 |
| Median age (IQR) | 48 (34.1–63.2) | 68.2 (58.9–77.2) | 50 (35–67) | 68 (58–76) |
| Male sex (%) | 7479 (38.8) | 2021 (52.2) | 9357 (37.4) | 1385 (52.6) |
| Race—Asian (%) | 590 (3.1) | 64 (1.7) | 2852 (11.4) | 225 (8.5) |
| Race—Black (%) | 3294 (17.1) | 802 (20.7) | 4011 (16) | 453 (17.2) |
| Race—White (%) | 5942 (30.8) | 1319 (34.1) | 9365 (37.4) | 1064 (40.4) |
| Ethnicity—Hispanic (%) | 5020 (26) | 1534 (39.6) | 3915 (15.6) | 464 (17.6) |
| Asthma (%) | 1816 (9.4) | 715 (18.5) | 1152 (4.6) | 345 (13.1) |
| Chronic kidney disease (%) | 2980 (15.4) | 2131 (55.1) | 3580 (14.3) | 1621 (61.5) |
| Chronic obstructive lung diseases (%) | 846 (4.4) | 545 (14.1) | 609 (2.4) | 283 (10.7) |
| Coronary artery disease (%) | 1253 (6.5) | 1243 (32.1) | 842 (3.4) | 627 (23.8) |
| Diabetes mellitus (%) | 2300 (11.9) | 1999 (51.6) | 1928 (7.7) | 1139 (43.2) |
| Diabetic nephropathy (%) | 266 (1.4) | 286 (7.4) | 217 (0.9) | 236 (9) |
| Diabetic neuropathy (%) | 325 (1.7) | 355 (9.2) | 141 (0.6) | 159 (6) |
| Diabetic retinopathy (%) | 135 (0.7) | 158 (4.1) | 93 (0.4) | 138 (5.2) |
| Diabetic vasculopathy (%) | 123 (0.6) | 193 (5) | 48 (0.2) | 70 (2.7) |
| Heart failure (%) | 1176 (6.1) | 1187 (30.7) | 676 (2.7) | 576 (21.9) |
| Hypertension (%) | 3986 (20.7) | 2928 (75.6) | 3592 (14.4) | 2227 (84.5) |
| Insulin use (%) | 1839 (9.5) | 1642 (42.4) | 910 (3.6) | 729 (27.7) |
| Myocardial infarction (%) | 499 (2.6) | 614 (15.9) | 375 (1.5) | 301 (11.4) |
| Obesity (%) | 3659 (19) | 1610 (41.6) | 3529 (14.1) | 1166 (44.3) |
| Proteinuria (%) | 3063 (15.9) | 1465 (37.8) | 1327 (5.3) | 617 (23.4) |
| HbA1c median (N) | 5.9 (2315) | 6.7 (1558) | 6.1 (895) | 6.7 (452) |
| Beta blocker (%) | 2208 (11.4) | 2198 (56.8) | 1746 (7) | 1406 (53.4) |
| Calcium channel blocker (%) | 1854 (9.6) | 1902 (49.1) | 1305 (5.2) | 1176 (44.6) |
| Thiazide/thiazide-like diuretic (%) | 664 (3.4) | 1345 (34.7) | 476 (1.9) | 816 (31) |
| Tested for SARS-CoV-2 infection (%) | 17812 (92.3) | 3500 (90.4) | 23578 (94.2) | 2464 (93.5) |
| Confirmed SARS-CoV-2 infection (%) | 4697 (24.3) | 1252 (32.3) | 2878 (11.5) | 479 (18.2) |
| SARS-CoV-2 infection/COVID-19 (%) | 7128 (36.9) | 1730 (44.7) | 4343 (17.4) | 652 (24.8) |
| Intubated/COV+ (% COV+) | 437 (6.1) | 242 (14) | 254 (5.8) | 74 (11.3) |
| Died/COV+ (% COV+) | 504 (7.1) | 261 (15.1) | 226 (5.2) | 69 (10.6) |

Breakdown of investigated cohort by ACEI/ARB exposure and site. Listed are all indications and comorbidities considered in the clinical analysis.

no detectable presence in normal donors (Fig. 6, Supplementary Fig. 11). Both of these enrichments represent unique cellular distributions in the COVID-19 patients relative to normal donors, and agrees with reports of immune cell-mediated influence of inflammation for COVID-19 patients.

**Depletion and enrichment of HLA types in COVID patients.** Finally, to probe host genetic diversity in widely expressed genes broadly implicated in immune responses, we also checked RNA data for potential enrichment or depletion of class I *HLA* type(s) among COVID patients. Using arcasHLA[24], we tallied frequencies of first-field locus-specific haplogroups (*serotype alleles*, in *HLA* parlance) called with default or greater confidence, among SARS-CoV-2 qRT-PCR-positive versus -negative patients who each had at least 100 class I *HLA*-mapped transcriptome reads. First-pass single-locus Fisher exact tests suggested modest depletion of *HLA-B*08* and/or *HLA-B*18*, and/or enrichment of *HLA-B*35*, *HLA-B*39*, and/or *HLA-B*48*, in COVID patients (Supplementary Fig. 12), but no such disparity in *HLA-A* or *HLA-C* (Supplementary Fig. 13). To more stringently buffer finite-sample dependence among calls at each locus, and the risk of spurious findings under multiple testing, we permuted qRT-PCR positive and negative labels, preserving sample diplotype calls in 100 randomly seeded runs of 1000 permutations each. Such permutation supported, by consistent extremity of original counts to those under permutation, putative COVID patient-depletion of *HLA-B*08* and *HLA-B*18* and COVID patient-enrichment of *HLA-B*39* and *HLA-B*48*, but not of *HLA-B*35* (Supplementary

Fig. 12B). We caution that these findings, while nominally significant, neglect plausible ancestry-differential COVID-19 infection incidence and/or ascertainment in our NYC cohort (which may strongly confound *HLA* frequency comparison), as well as intrinsic imprecision of diplotype inference from RNA data, linkage among class I *HLA* genes (and other major histocompatibility complex loci), and/or other sequence variation distinctive to specific subtypes of surveyed haplogroups.

## Discussion
In summary, these data showed that total RNA-sequencing data has great promise as a comprehensive and accurate host and pathogen profiling method. The RNA-seq data enabled a complete genetic map of the viruses, bacteria, host responses, and even HLA subtypes from the sample data source. Of note, a significant subset of our samples, including in nine qRT-PCR negative cases, had sufficient reads to assemble the SARS-CoV-2 genome de novo. Though these likely qRT-PCR false negatives could not be attributed to specific sequence changes (e.g., primer site mutations), their high frequency underscores the limitation of "gold standard" qRT-PCR approaches for SARS-CoV-2 detection. These results also highlight the need for open-source primer design, so these assays can be updated as a more granular picture of strain diversity and evolution is obtained through worldwide sequence efforts[25]. Nevertheless, the rate of SARS-CoV-2 assemblies or reads among qRT-PCR negative cases with ILI symptoms was low, despite a comprehensive search for these sequences among shotgun transcriptomics

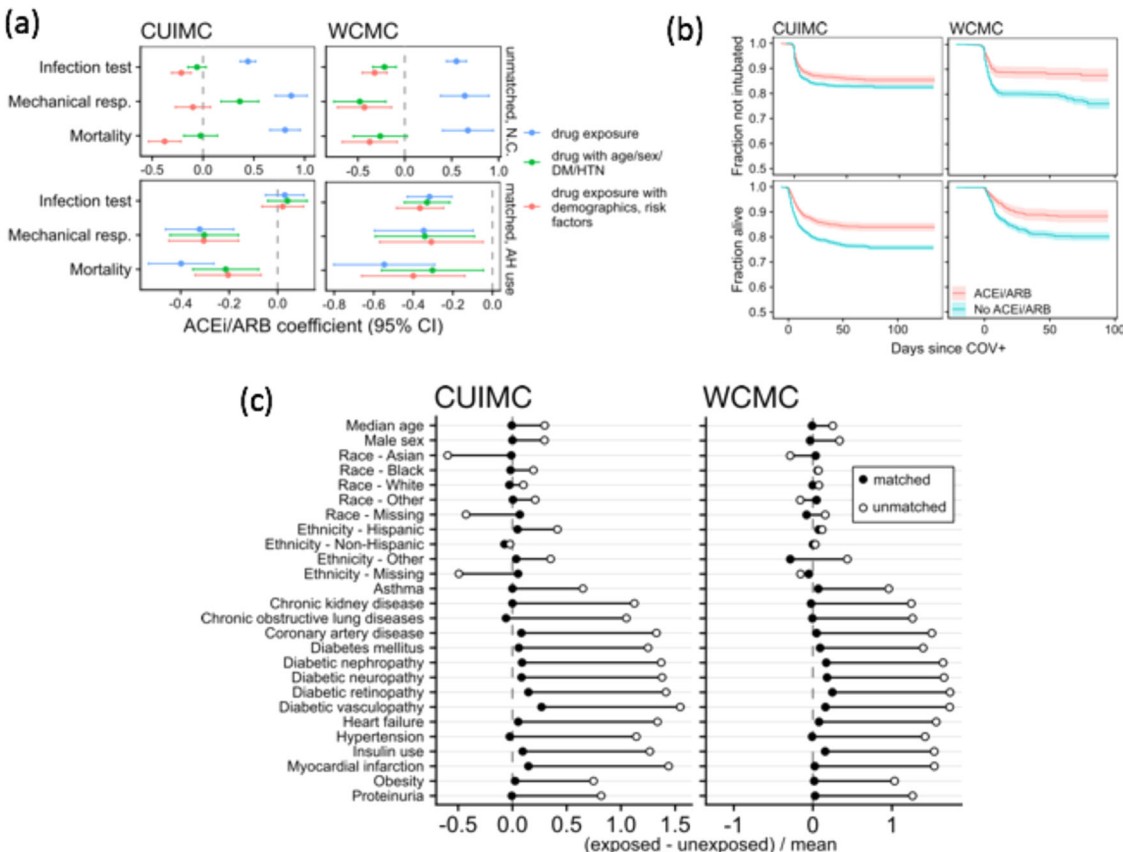

**Fig. 5 Host risk to SARS-CoV-2. a** Estimates for ACEI/ARB use effect on SARS-CoV-2 infection test result, intubation, and death. Upper panels show estimates from all patients without propensity matching (unmatched, no condition (N.C.)). Lower panels show estimates from propensity-matched patients with recent exposure to antihypertensive (AH) drugs (ACEI, ARB, BB, CCB, and THZ) (Supplementary Data 6). Center points show the estimated coefficient and the error bars show 95% confidence interval. Coefficients represent effect size estimates from logistic regression (for infection test outcome) or Cox proportional hazards regression (mechanical respiration and mortality), and can be interpreted as log odds ratios (former outcome) or log hazard ratios (latter outcomes). Colors indicate the variables used for regression on the cohort (drug exposure alone (blue), drug exposure plus age, sex, history of diabetes mellitus (DM), history of hypertension (HTN) (green), drug exposure plus all demographics and risk factors considered (red)). Left panels show estimates from NYP-CUIMC data, right panels show estimates from NYP-WCMC. **b** Curves for 50,821 patients requiring mechanical respiration (top panels) and survival (bottom panels), as a function of time since confirmed infection and recent ACEI/ARB exposure status. Cohorts created using propensity matching. Left panels give data from NYP-CUIMC, right panels give data from NYP-WCMC. **c** Comparison of cohorts before and after propensity matching. Each point represents a difference in the mean between exposed and unexposed cohorts, divided by the mean of exposed and unexposed cohorts. Filled circles give these standardized differences after propensity matching, unfilled circles give values before matching.

reads. These results indicate that existing standard detection methods are reasonably adapted to patterns of SARS-CoV-2 strain variation.

Notably, our phylogenetic analysis shows that the 20C clade comprises the majority of known NYC samples, including those sequenced outside of this study[3]. Though remaining NYC cases show a wide distribution across all identified clades, the predominance (>80% in NYC) of such a narrowly defined set of sequences within NYC from a minority (≤20%) Western European suggests either a founder effect or differential strain fitness. Though the Spike D614G mutation, which defines a superclade of 20C previously known as A2a[3], has been proposed to increase transmissibility (e.g., via higher viral load), and the low and stable prevalence of 20C in Western Europe since March suggests against differential fitness as an explanation for its dominance in NYC. A more likely explanation involves the early introduction and expansion of one or several 20C cases to NYC in late February, e.g., via the New Rochelle cluster. The gradual increases of 20C in the remainder of the USA post March may similarly reflect migration patterns of New Yorkers to the South and Midwest.

Our results further demonstrate the value of RNA-seq data by revealing distinct, host transcriptional programs that were activated during viral infection of the naso-/oropharynx with SARS-CoV-2. This includes upregulation of specific interferon pathway genes (*SHFL*, *IFI6*, *IFI27*, and *IFIT1*) that have been previously associated with the innate antiviral host immune response against other positive-strand RNA viruses (e.g., hepatitis C, Dengue virus). These results also provide clinical relevance for recently published results from animal and cellular models of SARS-CoV-2[19]. Our analyses also implicate expression perturbations of the ACE pathway in SARS-CoV-2 host response, including *ACE2*, and map these patterns from COVID-19 lung samples from autopsies[26]. Patients presenting with COVID-19 frequently harbor comorbidities such as hypertension, diabetes mellitus, and coronary heart diseases, all of which have been associated with increased disease severity[23,27]. Since these comorbidities are frequently treated with ACEIs and ARBs, one possibility is that these medications may make patients more susceptible to SARS-CoV-2 infection, but this has been limited in prior studies.

Here, we examined the risk of medication use and comorbidities in the context of ACEI or ARB in a large, retrospective

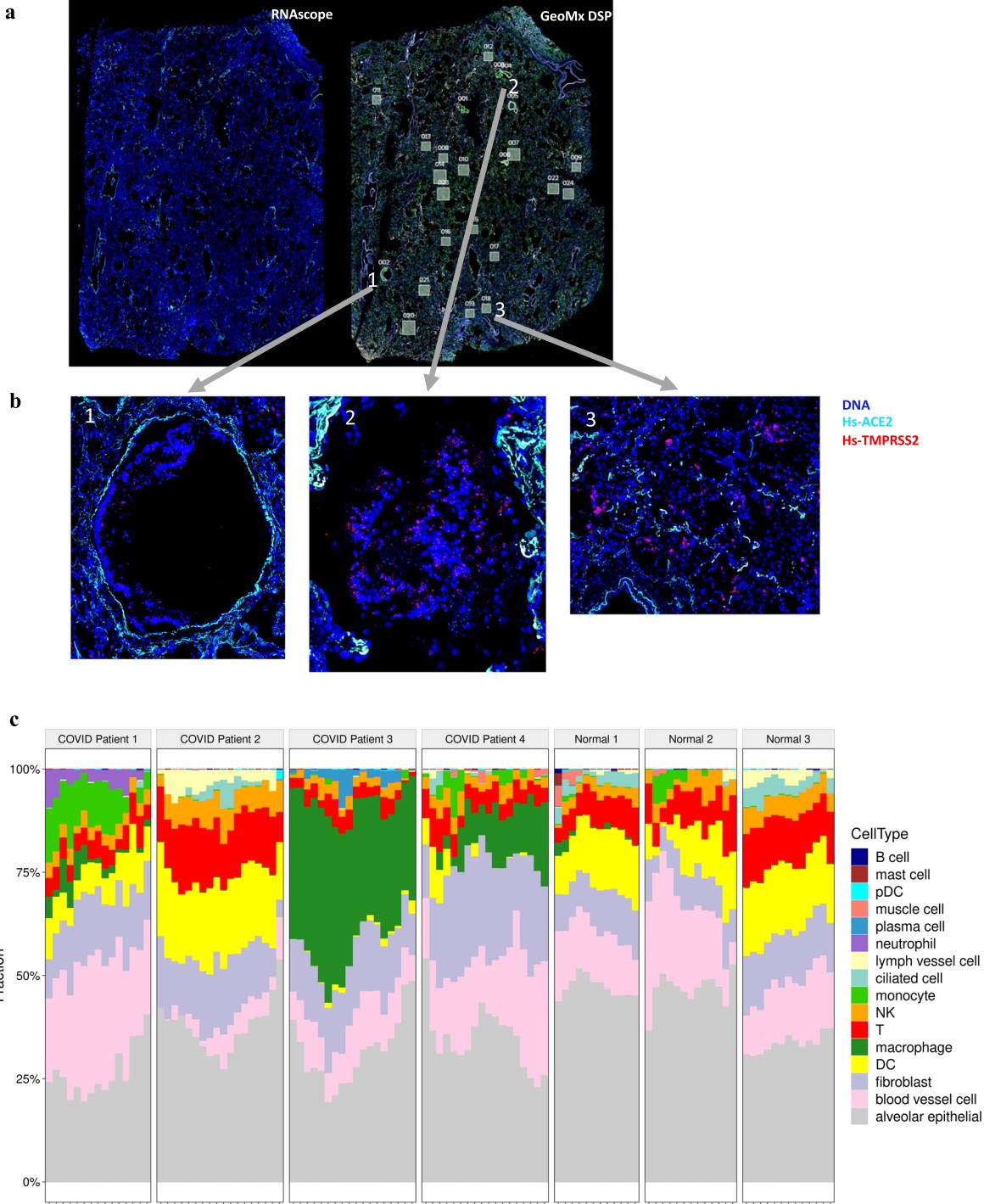

**Fig. 6 Spatial transcriptomic profiles of patient autopsies. a** Imaging from RNAscope showing fluorescence of target genes and imaging from GeoMx DSP defining regions of gene expression measurement for COVID Patient #4. **b** Close up of few selected regions of interest (ROI). **c** Cellular composition of 107 tissue regions. Each column shows the cell proportions within a single alveolar segment, as estimated from mixed cell deconvolution of GeoMx RNA data. Controls are from excess lung material from healthy lung transplant donors.

clinical analysis of 50,821 patient records. These data show that patients with ACEI/ARB exposure had lower risk of severe COVID-19 outcomes, after adjusting for demographics, indications, and other relevant comorbidities. A recent study reported a protective effect of ACEI/ARB exposure on mortality[28]. The associations we found support this finding, and we found similar associations for mechanical respiration requirements among infected patients. Further, our data did not show that other antihypertensive drug classes had any association with mechanical respiration requirements, though BB and CCB were associated with lower risk of death as well. While our data suggest a protective association between ACEI/ARB use and severe COVID-19 from two large New York hospitals (Columbia and Weill Cornell), our results are still preliminary, and caution should be taken in interpreting them. Because of the risk of residual confounding variables, prospective clinical trials would

be necessary before clinical guidelines should be changed. For example, if some patients are more susceptible because they already express high levels of ACE2, this could help with targeting the ACE pathway in these patients as a prophylactic method. However, if the cells respond to infection with ACE2 expression, and this leads to the cytokine storm seen in patients, then this could be used as a downstream treatment (post-infection), for when ACE2 interacts with TMPRSS2, such as the ongoing trials with camostat mesylate[16].

Mitigating a fast-spreading pandemic, such as COVID-19, requires scalable diagnostic and screening methods[29,30]. Such a situation calls for fast, scalable tests readily implemented at home or point-of-care. This one-tube, dual-primer colorimetric SARS-CoV-2 assay has been clinically-validated with qRT-PCR, capture, and total RNA-seq, and thus can help increase testing options; moreover it has recently become available in several commercial forms. Our preliminary results here and elsewhere demonstrate SARS-CoV-2 detection from oral specimens is feasible and even optimal[31,32]. As such, a LAMP-based approach on such sample types could allow facilities to increase testing capabilities by orders of magnitude. Viral pandemics can have significant, long-lasting detrimental impacts for affected countries, and thus it is crucial to deploy methods that can track and profile cases (e.g., RNA-seq, LAMP, qRT-PCR, capture) and provide a comprehensive view of host and viral biology[33]. These methods can help mitigate the medical and socioeconomic harm from viral outbreaks, as well as establish protective surveillance networks that can help defend against future pandemics.

## Methods

**Sample collection and processing.** Patient specimens were collected with patients' consent at New York-Presbyterian Hospital-Weill Cornell Medical Center (NYPH-WCMC) and then processed for qRT-PCR. Nasopharyngeal (NP) swab specimens were collected using the BD Universal Viral Transport Media system (Becton, Dickinson and Company, Franklin Lakes, NJ) from symptomatic patients. Samples were collected and processed through the Weill Cornell Medicine Institutional Review Board (IRB) Protocol 19-11021069. All relevant ethical regulations for work with human participants were complied with. Observational cohort analysis (ACEI/ARB) was done through the Columbia University IRB Protocol AAAL0601. Oropharyngeal samples data from University Hospital of Tuebingen were collected under IRB 243/2020A.

**Extraction of Viral RNA and qRT-PCR detection.** Total viral RNA was extracted from deactivated samples using automated nucleic acid extraction on the QIA-symphony and the DSP Virus/Pathogen Mini Kit (QIAGEN). One step reverse transcription to cDNA and real-time PCR (RT-PCR) amplification of viral targets, E (envelope) and S (spike) genes and internal control, was performed using the Rotor-Gene Q thermocycler (QIAGEN).

**Twist synthetic RNAs.** We used two fully synthetic RNAs made by in vitro transcription from Twist Biosciences, which were synthesized in 5 kb pieces with full viral genome coverage of SARS-CoV-2. They were sequence verified to ensure >99.9% viral genome coverage, and come as 1,000,000 copies per μL, 100 μL per tube. The two controls are from Wuhan, China (MN908947.3) and Melbourne, Australia (MT007544.1). Reference sequence designs came from NCBI: https://www.ncbi.nlm.nih.gov/nuccore/MT007544 and https://www.ncbi.nlm.nih.gov/nuccore/MN908947.3.

**Reverse transcriptase, quantitative real-time PCR (RT-PCR).** Clinical samples were extracted as described above and then tested with qRT-PCR using primers for the E (envelope) gene, which detects all members of the lineage B of the beta-CoVs, including all SARS, SARS-like, and SARS-related viruses, and a second primer set for the S (spike) gene, which specifically detects the SARS-CoV-2 virus. The reaction also contains an internal control serving as an extraction control and a control for PCR inhibition.

Samples were annotated using qRT-PCR cycle threshold (Ct) value for SARS-CoV-2 primers. Subjects with Ct less than or equal to 18 were assigned "high-viral load" label, Ct between 18 and 24 were assigned "medium viral load" and Ct between 24 and 40 were assigned "low viral load" classes, with anything above Ct of 40 was classified as negative. We also predicted a combined viral load score using Ct, GloMax QuantiFluor read-out from LAMP experiments and fraction of SARS-CoV-2 matching NGS reads in a sample. For this score (40-Ct), (LAMP read-out) and (log10(SARS-CoV-2 fraction + 1e − 6)) were all normalized between zero and one individually, and summed together using a combination weight of 5 for Ct, 3 for LAMP and 2 for NGS.

**LAMP primer sequences.** Primers (Supplementary Table 7) were designed using PrimerExplorer (v4.0), as per guidelines in[28] to find LAMP-compatible primers for the SARS-CoV-2 reference genome (NCBI). LAMP's inherent specificity (using 4–6 primers vs. 2 for PCR amplification), in combination with this in silico analysis, suggested little risk of cross-reactivity that might hinder specificity of N-gene primers sensitive for SARS-CoV-2 (Supplementary Data 5). Overall, chosen primers had less than 80% homology with the vast majority of tested pathogen sequences. For any organisms where a primer hit >80% homology, only one of the six primers had significant homology making an amplified product extremely unlikely. Overall, the results of this analysis predict no significant cross-reactivity or microbial interference. We also assessed the potential impact of sequence variation in circulating strains that might lead to poor amplification. In the thousands of sequences deposited in GISAID (Shu and McCauley, 2017), only one site in the priming region was observed to be polymorphic. The polymorphism (T30359C) was only observed in 106 of 6753 (<2%) sequences with coverage of this region. This variant overlaps the priming site of the LB primer but is not near a 3-prime end and is not anticipated to cause amplification failure. The data from Fig. 1 show the use of a single-tube, dual-primer protocol, wherein both the N2 gene and E-gene primers are present.

**The LAMP reaction setup.** For each well or Eppendorf tube, we used a set of six primers (above) for Gene N, the M1800 Colorimetric LAMP Master Mix (NEB), water, and 11.5 μL of the sample. The protocol is as follows:

1. Reagents added:

   a. 12.5 μL M1800 LAMP mix (NEB).
   b. 1–5 μL LAMP primers (Gene N or N2/E mix).
   c. 1–11.5 μL of sample.
   d. Remaining volume (to 25 μl) H$_2$O.

2. Vortex, spin down.
3. Place on Thermocycler at 65 °C for 30 min with lid at 105 °C.
4. Remove tubes, place on ice for 5 s.
5. Visualize over lab bench/ice/paper.

**Oropharyngeal lysate LAMP run.** Nasopharyngeal and/or oropharyngeal swab samples from 201 patients were collected (IRB 243/2020A Tuebingen) using a dry cotton swab (cliniswab DS, Aptaca Spa (Italy), #2170/SG/CS). Crude extraction was performed according to pending unpublished European patent [No. 20168 593.0]. In summary, the dry swab was transferred to a 15 ml falcon (Greiner Bio-one, #188.271), filled with 0.5 ml saline solution, and shaken vigorously for 30 min. Afterwards the liquid is transferred to a screw-cap (Sarstedt, #72.692.005). 10μl of the crude extract was added to 12.5 μl 2× NEB LAMP master mix (#M1800L), 2.5 μl of water, 1 μl 25× primer master mix Gen N, and was incubated at 65 °C for 30–40 min. A sample of a patient who tested positive using an approved qRT-PCR test (sample #1123) was used as an internal control. The read-out was performed visually by color change from pink to yellow or orange.

The RNA isolation was performed with 50 μl of the crude extract on the QIAsymphony with the DSP Virus/Pathogen Kit (Qiagen, #937055). The RT-PCR was performed using 5 μl of 85 μl eluate with TIB MolBiol Lightmix® MODULAR SARS AND WUHAN CoV E-Gene Kit. Analysis was done with the LightCycler(R) 480 II software and calculated CP values were used for statics and graphical analysis. The Standard curve with a synthetic RNA control (Twist Bioscience, #MT007544.1) was generated using the LAMP assay and in parallel with the qRT-PCR. The control RNA was diluted serially tenfold with water and the absolute copy number, ranging from $10^5$ to $10^{-5}$ was analyzed. Gel electrophoresis after visual read-out of the LAMP assay was done by loading 5 μl of the LAMP reaction with 5 μl 2× loading dye on a 1.5% agarose (Seakem LE Agarose, Lonza #50004) together with 5 μl of a 1 kB DNA Ladder (Roche). The electrophoresis was performed at 90 V constant.

The photometric read-out of the standard curve was performed in a plate reader. To this end the LAMP reaction was transferred into a 96-well V-shaped cell-culture plate (Greiner Bio-One, #651180). After measuring the absorbances at 432 and 560 nm, the relative absorbance abs (432)/abs (560) was calculated by subtraction of the negative (water) control from all samples (including the negative control). All values above a threshold of 0.1 are considered as a positive assay read-out and are marked with "+", all other values are negative and are marked "−". Statistical and graphical analysis were performed with GraphPad Prism 8.0.4.

**Light intensity and data processing.** Completed reactions were analyzed with the Promega GloMax Explorer (Promega GM3500) fluorometer using the QuantiFluor ONE dsDNA system (Promega E4871). This system recorded fluorometric read-out from each well using an emission filter of 500-550nm, an excitation filter set at

blue 475 nm, and a high sensitivity setting on the Glomax software. Values were then tabulated and compared with controls (positive and negative). The intensity threshold of 2.5× negative control was used as the threshold for positive detection.

**DNAse treatment, rRNA depletion, and RNAseq library construction**. For library preparation, all samples' total nucleic acid (TNA) were treated with DNAse 1(Zymo Research, Catalog # E1010), which cuts both double-stranded and single-stranded DNA. Post-DNAse digested samples were then put into the NEBNext rRNA depletion v2 (Human/Mouse/Rat), Ultra II Directional RNA (10 ng), and Unique Dual Index Primer Pairs were used following the vendor protocols from New England Biolabs (except for the first flowcell, see Supplementary Figures). Kits were supplied from a single manufacturer lot. Completed libraries were quantified by Qubit or equivalent and run on a Bioanalyzer or equivalent for size determination. Libraries were pooled and sent to the WCM Genomics Core or HudsonAlpha for final quantification by Qubit fluorometer (ThermoFisher Scientific), TapeStation 2200 (Agilent), and qRT-PCR using the Kapa Biosystems Illumina library quantification kit.

**Taxonomic classification of sequence data**. All complete genome or chromosome level assemblies from RefSeq database for archaea, bacteria, protozoa, fungi, human and viruses including SARS-CoV and SARS-CoV-2 genomes were downloaded and used for building a classification database for Kraken2 ($k = 35$, $\ell = 31$)[34,35].

To get an approximation for the positive and negative classification rate, the BBMap random-reads script was used to simulate 10 million 150 bp paired-end Illumina reads from the database sequences[36]. For the negative test all sequences in the database excluding SARS-CoV and SARS-CoV-2 genome were removed from the sequences and the simulated reads were mapped with the Kraken2 database (Supplementary Data 1).

For the positive test, the same process was repeated using only SARS-CoV-2 genome (Supplementary Data 2). Positive results show >99% of SARS-CoV-2 reads uniquely map to either SARS-CoV or SARS-CoV-2, with the remaining 1% are ambiguous, potentially matching multiple taxa (Supplementary Data 1, 2). All sequences were classified using the Kraken2 database. To remove the potential contamination of reads that are homologous across multiple species we used Kraken2 outputs to filter sequences to either human (uniquely matching Homo sapiens and no other taxon in our database), SARS-CoV-2 (either matching SARS-CoV or SARS-CoV-2 due to homology between these two viruses), and remaining reads that may be coming from unclassified, archaeal, bacterial, viral, fungal, protozoan or ambiguously mapping reads to human or SARS-CoV[37].

Using kraken2 classifications common respiratory pathogens were identified in clinical samples. Any SARS-CoV-2 negative sample with >0.01% relative abundance (normalized after the exclusion of any human, SARS-CoV-2 and uncharacterized reads) for presence of viral pathogens were classified as potential unrelated viral infection (Supplementary Fig. 9). These samples were used as controls during specific differential expression comparisons to compare the common effects of viral infections on host cells from SARS-CoV-2 infection.

**Viral genome assembly**. Reads unambiguously mapping to SARS-CoV or SARS-CoV-2 were aligned to the Wuhan-Hu-1 (Genbank accession MN908947.3) reference using bwa mem[38]. Variants were called using iVar, and pileups and consensus sequences were generated using samtools[39–41]. Any sample with >99% coverage above 10× depth for SARS-CoV-2 genome were taken as reliable samples, which resulted in 155 samples (146 positive, 9 negative). 155 clinical samples were compared to 46,581 SARS-CoV-2 sequences from GISAID (as of June 16, 2020)[42,43]. All sequence filtering, alignments, phylogenetic inference, temporal ordering of sequences, and geographic reconstruction of likely transmission events were done using Nextstrain[7,44,45]. Nextstrain was configured to generate a New York State-focused build. Sequence identity and coverage metrics were calculated using Mview[46]. Phylogenetic trees were created using Nextstrain's augur as described above, and visualized using the ggtree package in R[47].

**Viral variant calling and allelic fraction estimation**. Full-length viral consensus sequences were aligned to the Wuhan-Hu-1 reference using bwa mem[38] with default settings. Variants were identified by enumerating the coordinates and query/reference subsequences associated with mismatches (SNV) and gaps in the query (deletion) and reference (insertions) using R/Bioconductor (GenomicRanges, Rsamtools, Biostrings packages) and gChain (https://github.com/mskilab/gChain) packages. Exhaustive variant calling on read alignments was additionally performed using bcftools mpileup and call, with variant read support (VAF, alternate allele count) enumerated with the R/Bioconductor Rsamtools package.

For each variant, a posterior distribution of VAF was computed using a beta distribution with shape parameters $\theta$ comprising reference and alternate allele counts and pseudo count of 0.5. Variants were classified as het (heterogenous) if $P(VAF > 0.05 \land VAF < 0.95|\theta) > 0.95$. For a given specimen, posterior VAF distributions of $k$ heterogenous variants were then integrated using a histogram density estimator by summing the posterior VAF density across all variants at each point of a grid of 100 points evenly spaced in the $(0, 1)$ line. This (unnormalized) mixture density was visualized alongside the individual VAF densities as an estimate of the probability density of a putative viral subclone.

**Cell deconvolution analysis**. Bulk RNAseq data were deconvolved into cell composition matrices by MUSIC algorithm[48], via reference single-cell RNAseq data from upper respiratory epithelium obtained from nasal brushes and upper airway and lung cells[49].

**Human transcriptome analysis**. The reads that mapped unambiguously to the human reference genome via Kraken2 were used to detect the host transcriptional response to the virus. Reads matching *Homo sapiens* were trimmed with Trim-Galore, aligned with STAR (v2.6.1d) to the human reference build GRCh38 and the GENCODE v33 transcriptome reference, gene expression was quantified using feature Counts, stringTie and salmon using the nf-core RNAseq pipeline[50–55]. Sample QC was reported using fastqc, RSeQC, qualimap, dupradar, Preseq, and MultiQC[56–60]. Samples that had more than 10 million human mapped reads were used for differential expression analysis. Reads, as reported by feature Counts, were normalized using variance-stabilizing transform (vst) in DESeq2 package in R for visualization purposes in log-scale[61]. Limma voom and DESeq2 were used to call differential expression with either Positive cases vs. Negative, or viral load (High/Medium/Low/None excluding any samples with evidence of other viral infections) as reported by either qRT-PCR cycle threshold (Ct) values, or using the inverted normalized Ct value as continuous response for viral levels, where Ct of 15 is 1.0 and Ct of >40 is taken as 0[62]. Genes with BH-adjusted $p$ value < 0.01 and absolute log2 fold-change greater than 0.58 (at least 50% change in either direction) were taken as significantly differentially regulated[19]. The same approaches were repeated correcting for potential confounders in our data in two ways. In the first correction ciliated cell fraction (as predicted by MUSIC) was added as another covariate to our model. For the second correction SVA was run on the data and the resulting two surrogate variables were included in a multivariate model[63]. The complete gene list for all comparisons are given in Supp Data 3. Resulting gene sets were ranked using log2 fold-change values within each comparison and put into GSEA (Supplementary Data 4) to calculate gene set enrichment for molecular signatures database (MSigDB), MGI Mammalian Phenotypes database and ENCODE transcription factor binding sets[64–68]. Any signature with adjusted $p$ value < 0.01 and absolute normalized enrichment score (NES) ≥ 1.5 were reported (Supplementary Data 3).

**Cross-reactivity analysis**. Primers were compared with a list of sequences from organism from the same genetic family as SARS-CoV-2 and other high-priority organisms listed in the United States Food and Drug Administration's Emergency Use Authorization Template (https://www.fda.gov/media/135900/download). Using the sequence names in the EUA template, the NCBI taxonomy database was queried to find the highest quality representative sequences for more detailed analysis. Primers were compared to this database using Blast 2.8.1 and the following parameters (word size: 7, match score: 2, mismatch score: −3, gap open cost: 5, gap extend cost: 2). Up to 1000 hits with $e$ value > 10 were reported.

**Inclusivity analysis**. Unique, full-length, human-sample sequences were downloaded from the GISAID web interface. These sequences were aligned to NC_045512.2 (Wuhan SARS-CoV-2) using minimap2 -x asm5 and visually inspected using IGV 2.8.0 with allele frequency threshold set to 0.01.

**ACE inhibitor/angiotensin receptor blocker clinical cohort analysis**. We estimated the effects of ACEI/ARB exposure on COVID-19 using an observational case-control analysis of electronic health record data from two NYC hospitals (CUIMC and WCMC). Due to legal and institutional restrictions on data sharing, we performed the analysis at each site separately. We used data from clinical encounters occurring between March 10 and June 15, 2020. Our study included all individuals who received a valid SARS-CoV-2 swab test. We used two cohorts for comparison[22]: all individuals who received a positive SARS-CoV-2 swab test result and[21] all individuals from the first cohort who were exposed to any of four anti-hypertensive drug classes (ACEI/ARB, BB, CCB, and THZ). A patient was considered exposed to one of these drug classes if they received and filled a prescription for a drug in that class at any time in 2020.

We considered two severe COVID-19 outcomes: intubation and death following confirmed infection. Individuals with a single positive test result were considered to have been infected and were included in our analysis. Intubation or death following confirmed infection was considered related to COVID-19. As we cannot determine from our data the date of infection, an individual became at risk of intubation and death due to COVID-19 7 days before their first positive test result or their first COVID-19 diagnosis. We determined mortality from a death note filed by a resident or primary provider that records the date and time of death. Intubation was used as an intermediary endpoint and is a proxy for a patient requiring mechanical respiration. We used note types that were developed for patients with SARS-CoV-2 infection to record that this procedure was completed. In addition, we validated outcome data derived from notes against the patient's medical record using manual review. As a comparison, we also performed a logistic regression using the same approach for the outcome of the infection test result itself using the full cohort of tested individuals.

We gathered data on the following 19 covariates, which were used both for propensity matching and for covariate adjustment: age, sex, race, ethnicity,

hypertension, obesity, chronic kidney disease, coronary artery disease, heart failure, myocardial infarction, diabetes mellitus, diabetic nephropathy, diabetic neuropathy, diabetic retinopathy, diabetic vasculopathy, asthma, chronic obstructive lung diseases, proteinuria, and insulin use. Individuals without demographic information (first four covariates) were excluded from the analysis. The final 15 variables represent disease history and are defined in detail in the supplement.

**Experimental statistical methods.** Using a combination of two cohorts and two severe outcomes, we made four propensity-matched, adjusted estimates of effect for ACEI/ARB. For comparison, we performed the same analysis using three other antihypertensive classes (BB, CCB, THZ). To inspect the degree of confounding due to covariates, we also estimated each effect with and without propensity matching and three levels of covariate adjustment. We used a Cox proportional hazards model for both outcomes. Propensity matching used logistic regression with the above 19 covariates using a 3:1 ratio of controls to cases in each of 100 propensity score bins, selecting none when the bin did not contain both cases and controls.

**Statistical and visualization software.** All electronic health data analyses were performed in Python 3.7 and all models were fit using R 3.6.3. Survival analyses (Cox regressions and survival curves) were performed with the survival package for R, version 3.1-12. Additional statistical analyses, processing, transformation, and visualization of genomic data were completed in R/Bioconductor ("Rsamtools", "GenomicRanges", and "Biostrings") and additional Imielinski Lab R packages ("gTrack", "gChain", "gUtils", and "RSeqLib") available at https://github.com/mskilab. Visualization of phylogenies was completed using Auspice and the 'ggtree' and 'ape' libraries for R.

**The sequential organ failure assessment (SOFA).** The SOFA score is a severity of illness score that sums six separate organ dysfunction subscores, was used to characterize the burden of organ dysfunction. For the central nervous system, kidney, liver, and coagulation organ dysfunction subscores, traditional SOFA methodology[69] was used, based on patient status in the first 24 h following admission to the hospital (can be whatever time frame that you used). The respiratory SOFA subscore requires the selection of the lowest PaO2/FiO2 of the 24 h period and associates a lower ratio with a more severe dysfunction score, in situations outside of an ICU setting a PaO2 is often not available. We used a commonly accepted imputation technique to impute a PaO2 from a SpO2[70]. The cardiovascular SOFA subscore was updated with additional vasopressors according to a norepinephrine equivalency table from a clinical trial[71]. Specifically, phenylephrine and vasopressin were converted to a norepinephrine equivalency. Missing data for each subscore was treated as normal.

*Spatial gene expression.* RNA from FFPE lung sections was extracted with the RNeasy FFPE Kit (Qiagen). Only the samples with A260/A280 ratios of ≥1.6 were used. At least 100 nanograms of RNA was loaded for hybridization with the nCounter® PanCancer IO360™ Gene Expression Panel spiked with the COVID-19 panel, according to the manufacturer's instructions and quantified by the nCounter® MAX Analysis System (NanoString Technologies, Seattle, WA, USA). Transcript counts were normalized to ERCC positive controls and housekeeper reference gene expression prior to analysis.

*RNA/NGS Slide Preparation for GeoMx DSP.* For GeoMx DSP slide preparation, we followed GeoMx DSP slide prep user manual (MAN-10087-04). Briefly, tissue slides were baked in a drying oven at 60 °C for 1 h and then loaded to Leica Biosystems BOND RX FFPE for deparaffinization and rehydration. After the target retrieval step, tissues werewere treated with Proteinase K solution to expose RNA targets followed by fixation with 10% NBF. After all tissue pre-treatments were done, tissue slides were unloaded from the Leica Biosystems BOND RX and incubated with RNA probe mix (CTA and COVID-19 spike-in panel) overnight. The next day, tissues were washed and stained with tissue visualization markers; CD68-647 at 1:400 (Novus Bio, NBP2-34736AF647), CD3-594 at 1:400 (Abcam, ab196147), CD45-594 at 1:10 (NanoString Technologies), PanCK-532 at 1:20 (NanoString Technologies) and/or SYTO 13 at 1:10 (Thermo Scientific S7575).

*GeoMx DSP sample collections.* For GeoMx DSP sample collections, we followed GeoMx DSP instrument user manual (MAN-10088-03). Briefly, tissue slides were loaded to GeoMx DSP instrument and then scanned to visualize whole tissue images. For scanning tissues, we used 100 ms for imaging DNA and 400 ms for all the other visualization markers. After whole tissue images stitched, we applied customized render setting to each tissue since tissue background and target protein expressions are highly sample-dependent. Roughly, the min and max of the render setting is from 20 to 200 and 1000 to 15,000, respectively. For each tissue sample, we collected four types of functional tissue regions; vascular zone, large airway, alveoli zone, and macrophages. Each tissue region was carefully selected by a board-certified pathology and then segmented with corresponding fluorescent tissue markers. Twenty-four to twenty-three GeoMx DSP samples were selected per tissue.

**RNAscope assay and imaging.** To visualize viral loads, we performed RNAscope assays using V-nCoV2019-S (Opal 570), Hs-ACE2 (Opal 620) and Hs-TMPRSS2 (Cy5) probes. We used serial tissue sections of the ones used for GeoMx DSP for sample preparations and followed the standard manufacture's protocol. We used GeoMx DSP to scan tissues using 100msec for imaging DNA (Syto13) and 200msec for all the other channels. We applied the following min and mas to the stitched images; DNA: 20 and 8,000, V-nCoV2019-S (Opal 570) and ACE2 (Opal 620): 200 and 800 and Hs-TMPRSS2 (Cy5): 500 and 3,000.

**GeoMx DSP NGS library preparation and sequencing.** Each GeoMx DSP sample was uniquely indexed using Illumina's i5 × i7 dual-indexing system. Four uL of a GeoMx DSP sample was used in a PCR reaction with 1 μM of i5 primer, 1 μM i7 primer, and 1X NSTG PCR Master Mix. Thermocycler conditions were 37 °C for 30 min, 50 °C for 10 min, 95 °C for 3 min, 18 cycles of 95 °C for 15 s, 65 °C for 60 s, 68 °C for 30 s, and final extension of 68 °C for 5 min. PCR reactions were purified with two rounds of AMPure XP beads (Beckman Coulter) at 1.2× bead-to-sample ratio. Libraries were paired-end sequenced (2 × 75) on a NextSeq 550 up to 400M total aligned reads.

**Normalization and removal of gene outliers.** For each sample, both a CTA-specific and a Covid-19 spike-in-specific negative probe normalization factor was generated based on the geometric mean of negative probes in each pool. Probes for each gene were screen for outliers and their raw counts collapsed to a single estimate of counts using their geometric mean. These gene count estimates were then divided by their respective negative normalization factor to normalize the data. For the Differential expression analysis, focus was given to the 207 Alveolar AOIs because they contained the bulk of the data. Of the 1837 genes, 1796 passed our criterion of limit of quantification (LOQ > 2).

**Dimensional reduction.** Dimensional reduction was performed to visualize large trends and clusters across the genes. To do this, the FactoMineR R package[72] was used for principal component analysis and the package Rtsne[73] was used to generate t-Distributed Stochastic Neighbor Embedding (tSNE) clustering (perplexity set to 30).

**Deconvolution of cell proportions.** Cell mixing proportions were estimated using the InSituSort R library, which performs mixture deconvolution using constrained log-normal regression. The algorithm was run using a cell profile matrix derived from the Human Cell Atlas adult lung 10X dataset and appended with a neutrophil profile derived from snRNA-seq of lung tumors[74].

**Differential expression analysis.** Differential expression analysis was performed by fitting each gene's normalized log2 expression level using a Linear Mixed Effect model with the R package lmerTest[29]. Different DE contrasts were made with Covid-19 samples as "baseline" to compare genes that are upregulated or down-regulated compared to Flu, non-viral ARDS, or normal alveolar lung. Because one Covid-19 sample appeared to have higher viral load compared to others (see results), we ran DE analysis with and without this individual (15 Alveolar AOIs in total). In all cases, we treated "Disease" status (Covid-19 vs. normal) as a level-2 fixed effect and accounted for within patient multiple sampling by treating patient ID as a random effect. Disease was allowed to have varying intercepts and slopes. Satterthwaite's approximation for degrees of freedom for $P$ value calculation was used[29]. The estimate of "Disease" from the Mixed Effect model is the log2 fold change. Hierarchical clustering of the top 30 DE genes was performed with the R package pheatmap.

*Gene set enrichment analysis.* Following differential expression on all normalized features, we used Gene Set Enrichment Analysis (GSEA) on the ranked expression values using the GSEA and Molecular Signature Database (MSigDB[68]).

**Differential expression validation from UCSF data.** Following sequencing of sample libraries, quality control was performed on the fastq files to ensure the sequencing reads met pre-established cutoffs for number of reads and quality using FastQC (version 0.11.8)[75] and MultiQC (version 1.8)[57]. Quality filtering and adapter trimming were performed using BBduk tools (version 38.76, https://sourceforce.net/projects/bbmap). Remaining reads were aligned to the ENSEMBL GRCh38 human reference genome assembly (Release 33) using STAR (version 2.7.0f)[76] and gene frequencies were counted using feature Counts (version 2.0.0) within the Subread package[77]. Comparative analysis of DEGs was performed using a generalized linear model (GLM) implemented in the edgeR Bioconductor package (version 3.30.3)[54], using a Benjamini–Hochberg corrected $p$ value of <0.01 (Supplementary Data 3).

Hierarchical clustering of DEGs was performed in R (version 4.0.0) using the ComplexHeatmap and pheatmap package, and figures were produced using the ggplot2 package. For NP and WB, the top 100 DEGs sorted by $p$ value with a Bonferroni corrected $p$ value of <0.001and <0.01 respectively were included. For the comparison between hospitalized and outpatients, all the DEGs with a Bonferroni corrected $p$ value of <0.01 were included. Clustering was performed based on Euclidean distance with complete linkage, after exclusion of noncoding genes.

Signaling pathway analyses and heatmaps were generated using Ingenuity Pathway Analysis (IPA) software (Qiagen)[78]. The molecule activity predictor tool of IPA was used to predict gene upregulation or downregulation and pathway activation or inhibition. The enrichment score $p$ value was used to evaluate the significance of the overlap between predicted and observed genes, while the z-score was used to assess the match between observed and predicted regulation or downregulation.

Classifiers were developed using scikit-learn (version 1.2.2)[79] in Python. Several different classifier models were evaluated in parallel and the one with optimal performance on the training data was selected. These candidate classifier models included a Linear Support Vector Machine, Linear Discriminant Analysis, and a Deep Neural Network, all within the scikit-learn package. Reduced gene panels were selected using Lasso[80] and a customized reverse search across the resulting feature set. This search iteratively removed the remaining gene with the lowest significance as measured by its Lasso coefficient, performed classifier training, and reported sensitivity, specificity, and accuracy across the training set. These results were then manually reviewed to balance each of them with a priority placed on specificity and number of genes. Receiver operating characteristic (ROC) curves were generated using pROC package in R[81].

**HLA analysis**. We mapped RNA reads to human reference genome GRCh38 using STAR v2.7.3a, and called likely *HLA* diplotypes for each sample by arcasHLA (Docker image quay.io/chai/arcas_hla:0.0.1, as pulled from master branch of https://github.com/RabadanLab/arcasHLA on 8 June 2020).

**Hybrid capture-base next-generation sequencing**. Five positive and one negative clinical samples were used to compare LAMP results with RT-qPCR and a hybrid capture-based NGS assay. RNA was isolated and purified in duplicates (250 μl input) using the Direct-zol DNA/RNA MiniPrep kit (Zymo). Viral transport media was used as negative extraction control. Extracted and purified RNA samples were converted to cDNA through random priming using Random Primer 6, ProtoScript II First Strand cDNA Synthesis Kit, and NEBNext Ultra II Non-Directional RNA Second Strand Synthesis kit (New England BioLabs).

The cDNA samples were converted to Illumina TruSeq-compatible libraries with Twist's Library Preparation Kit with Unique Dual Indices (Twist Bioscience). After cDNA library generation, samples were pooled in 8-plex reactions using 187.5 ng of each barcoded library. Hybridization reactions were performed for 2 h using Twist Fast Hybridization reagents and SARS-CoV-2 Research Panel, a biotin-bound panel that targets libraries containing the SARS-CoV-2 virus. Hybrid capture libraries were spiked with 1% PhiX and sequenced on an Illumina NextSeq 550 platform using a NextSeq500/550 High Output kit (Illumina) set to 150bp single end reads.

Extracted RNA was also tested using RT-qPCR (GenArraytion COVID-19 duplex RT-qPCR) to get CT values, and LAMP. Hybrid capture NGS data were analyzed using the COVID-DX Software (Biotia Inc.), optimized for the SARS-CoV-2 NGS Assay. The software includes removal of low-quality reads, alignment to the SARS-CoV-2 genome (NC_045512.2), and elimination of off-target reads by alignment to other genomes including human (GrC38) and 26 additional microbial genomes. Presence or absence of SARS-CoV-2 was determined by calculating coverage of NC_045512.2 at 1× depth using a sliding window analysis (window size 1000 and step size 100) and determining the integral under the curve. After log transformation, we used a cutoff of 8.6 (for samples with <10,000 bases on target), and 9.6 (for samples with >10,000 bases on target) to call presence or absence. The pipeline also calls germline variants that differ from the reference genome (NC_045512.2) and estimates viral titer using a multivariate model of coverage and evenness across the viral genome[82–108].

**Ethical approval**. Tissue samples were provided by the Weill Cornell Medicine Department of Pathology. The Tissue Procurement Facility operates under IRB approved protocol and follows guidelines set by Health Insurance Portability and Accountability Act (HIPAA). Experiments using samples from human subjects were conducted in accordance with local regulations and with the approval of the IRB at the Weill Cornell Medicine. The autopsy samples were collected under IRB protocols 20-04021814 and 19-11021069. Autopsy consent is provided on all cases by the next of kin. This consent includes use of tissue for research. Observational cohort analysis (ACEI/ARB) was done through the Columbia University IRB Protocol AAAL0601 and the Weill Cornell Medicine IRB protocol 20-04021820. Oropharyngeal samples data from University Hospital of Tuebingen were collected under IRB 243/2020A.

**Reporting summary**. Further information on research design is available in the Nature Research Reporting Summary linked to this article.

## Data availability
All raw sequence files (FASTQs) and metadata for specimens, including per-run metrics and QC data, have been submitted to the database of Genotypes and Phenotypes dbGAP (accession #38851 and ID phs002258.v1.p1). A total of 155 complete viral sequences were assembled from these data. The GenBank accession IDs for the viral genomes are MW493710-MW493863. Source data are provided with this paper.

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

## Acknowledgements

We thank New York-Presbyterian Hospital, the Clinical Translational Science Center (ULI TR000457), the Joint Clinical Trials Office, the Core Facilities at Weill Cornell Medicine, the Clinical Laboratories at New York-Presbyterian Hospital, the Scientific Computing Unit (SCU), OneCodex, the XSEDE Supercomputing Resources and the GISAID Initiative curators and submitters (Supp Data 7). We also thank New England Biolabs for providing the reagents for preliminary testing of the LAMP protocols, as well as Eileen Dimalanta and Ted Davis for technical discussions. We would like to thank Y.K. from Nanostring who supported the spatial transcriptomics analysis. The authors wish to thank the following members of the HudsonAlpha Discovery team who supported the RNASeq experiments described in the paper: Colleen Cowan, John Mote, Arianna Pionzio, Melanie Robinson, and Madison Robison. We thank AWS for sponsoring storage on DNAnexus. We are grateful for support from Cynthia Polsky, the STARR Foundation (I13-0052) the Vallee Foundation, the WorldQuant Foundation, The Pershing Square Sohn Cancer Research Alliance, Citadel, the National Institutes of Health (R01MH117406, R25EB020393, R01AI151059), the Bill and Melinda Gates Foundation (OPP1151054), the NSF (1840275), the National Center for Advancing Translational Sciences of the National Institutes of Health (UL1TR000457, CTSC), the Intramural Research Program of the National Library of Medicine, NIH, and the Alfred P. Sloan Foundation (G-2015-13964). F.J.S. supported by the National Institute of Allergy and Infectious Diseases (1U19AI144297-01). M.Z. supported by T15LM007079. NPT and UOG supported by R35GM131905. N.A.I. was supported by the National Center for Advancing Translational Sciences of the National Institutes of Health under Award Number TL1TR002386 and DB, DP and TI were supported by a grant from the BadenWürttemberg Ministery of Science and Art. IH is supported by 1R35GM138152-01.

## Author contributions

C.E.M. led the study design and coordination and H.R. led the clinical collection and validation work in the NYP CLIA laboratory, as well as M.I. Overall supervision and protocol development and implementation for the Zymo RNAClean and NEB assays (S.L.). D.B. and C.M.Z. performed the LAMP experiments to validate the assay, established a method to quantify LAMP output, and developed a protocol for clinical use of the assay. C.W., D.X., P.R., J.G., L.C., and J.X. assisted with sample preparation and sequencing. D.B., A.I., D.P., and T.I. developed and validated the testing from raw oropharyngeal swabs. C.E.M., D.D., J.F., M.I., A.S., J.R., M.M., E.A., I.H., D.M., D.T.M., J.T.M., B.W.L., M.Z., U.G., N.P.T., N.A.I., A.F., C.S.C., F.S., N.I., M.S., D.P., M.Z.,V.R., E.S., E.Sc., C.H., A.G., Y.S., V.S., S.F., A.F., C.S.C., N.P., and N.T. performed analyses. D.D., C.Mz., N.A.I., M.S., B.Y., K.R., C.B. coordinated and collected samples. E.A. submitted the IRB application and helped with clinical coordination. E.A., S.L., M.I., M.S., L.F.W., M.L., M.C., H.R., K.R., C.S.C., N.M.P., B.L., N.T., Y.K., J.R., T.H., S.W., M.B., D.M., M.C.R., D.N.S., J.B., H.W., N.O.H., J.R., Y.C., P.A.S., A.S., J.T.M., D.T.M., A.I., E.S., T.R.C., J.S., A.C., P.V., A.M., S.S., I.H., A.B., T.I., M.C., S.W., C.C., N.P.T., U.O.G, and R.S. helped plan, refine, and interpret analyses. All authors reviewed, edited, and approved the paper.

## Competing interests

N.T. and B.W.L. are employees at New England Biolabs. R.E.S. is on the scientific advisory board of Miromatrix Inc. Biotia employees and advisors include M.C.R., D.N.S., J.B., H.W., C.E.M., and N.O. D.B. is a co-founder of Poppy Inc; A.F and C.S.C are employees of DNAnexus. Authors not listed here do not have competing interests.

## Additional information

Daniel Butler [1,34], Christopher Mozsary[1,34], Cem Meydan [1,2,3,34], Jonathan Foox[1,2,34], Joel Rosiene[4,5,34], Alon Shaiber[4,5,6,34], David Danko[1,34], Ebrahim Afshinnekoo[1,2,3], Matthew MacKay[1], Fritz J. Sedlazeck [7], Nikolay A. Ivanov [1,2,8], Maria Sierra[1,2], Diana Pohle[9], Michael Zietz [10], Undina Gisladottir [10], Vijendra Ramlall[10,11], Evan T. Sholle [12], Edward J. Schenck[13], Craig D. Westover[1], Ciaran Hassan[1], Krista Ryon[1], Benjamin Young[1], Chandrima Bhattacharya[1], Dianna L. Ng[14], Andrea C. Granados[14,15], Yale A. Santos[14,15], Venice Servellita[14,15], Scot Federman [14,15], Phyllis Ruggiero[5], Arkarachai Fungtammasan[16], Chen-Shan Chin[16], Nathaniel M. Pearson[17], Bradley W. Langhorst [18], Nathan A. Tanner[18], Youngmi Kim[19], Jason W. Reeves[19], Tyler D. Hether[19], Sarah E. Warren[19], Michael Bailey[19], Justyna Gawrys[5], Dmitry Meleshko[1,20], Dong Xu[21], Mara Couto-Rodriguez [22], Dorottya Nagy-Szakal[22,23], Joseph Barrows[22], Heather Wells[22], Niamh B. O'Hara[22,23], Jeffrey A. Rosenfeld[24,25], Ying Chen [24], Peter A. D. Steel[26], Amos J. Shemesh [26], Jenny Xiang[21], Jean Thierry-Mieg[27], Danielle Thierry-Mieg[27], Angelika Iftner[9], Daniela Bezdan [9], Elizabeth Sanchez[13], Thomas R. Campion Jr. [12,28], John Sipley[5], Lin Cong[5], Arryn Craney[5], Priya Velu[5], Ari M. Melnick [13], Sagi Shapira [10], Iman Hajirasouliha[1,2,6], Alain Borczuk [13], Thomas Iftner[9], Mirella Salvatore [13,28], Massimo Loda[5], Lars F. Westblade[5,13], Melissa Cushing[5], Shixiu Wu[29,30], Shawn Levy [31], Charles Chiu [14,15,32], Robert E. Schwartz [13✉], Nicholas Tatonetti [10✉], Hanna Rennert [5✉], Marcin Imielinski [4,5,6✉] & Christopher E. Mason [1,2,3,33✉]

[1]Department of Physiology and Biophysics, Weill Cornell Medicine, New York, NY, USA. [2]The HRH Prince Alwaleed Bin Talal Bin Abdulaziz Alsaud Institute for Computational Biomedicine, Weill Cornell Medicine, New York, NY, USA. [3]WorldQuant Initiative for Quantitative Prediction, Weill Cornell Medicine, New York, NY, USA. [4]New York Genome Center, New York, NY, USA. [5]Department of Pathology and Laboratory Medicine, Weill Cornell Medicine, New York, NY, USA. [6]Englander Institute for Precision Medicine and the Meyer Cancer Center, Weill Cornell Medicine, New York, NY, USA. [7]Human Genome Sequencing Center, Baylor College of Medicine, Houston, TX, USA. [8]Clinical & Translational Science Center, Weill Cornell Medicine, New York, NY, USA. [9]Institute of Medical Virology and Epidemiology of Viral Diseases, University Hospital Tuebingen, Tuebingen, Germany. [10]Department of Biomedical Informatics, Department of Systems Biology, Department of Medicine, Institute for Genomic Medicine, Columbia University, Columbia, NY, USA. [11]Department of Cellular, Molecular Physiology & Biophysics, Columbia University, Columbia, NY, USA. [12]Information Technologies & Services Department, Weill Cornell Medicine, New York, NY, USA. [13]Department of Medicine, Weill Cornell Medicine, New York, NY, USA. [14]Department of Laboratory Medicine, University of California, San Francisco, CA, USA. [15]UCSF—Abbott Viral Diagnostics and Discovery Center, San Francisco, CA, USA. [16]DNAnexus, Inc., Mountain View, CA, USA. [17]Root Deep Insight, Boston, MA, USA. [18]New England Biolabs, Ipswich, MA, USA. [19]NanoString Technologies, Seattle, WA, USA. [20]Tri-Institutional Computational Biology & Medicine Program, Weill Cornell Medicine, New York, NY, USA. [21]Genomics Resources Core Facility, Weill Cornell Medicine, New York, NY, USA. [22]Biotia, Inc., New York, NY, USA. [23]Department of Cell Biology, SUNY Downstate Health Sciences University, New York, NY, USA. [24]Rutgers Cancer Institute of New Jersey, New York, NJ, USA. [25]Department of Pathology, Robert Wood Johnson Medical School, New York, NJ, USA. [26]Department of Emergency Medicine, Weill Cornell Medicine, New York, NY, USA. [27]National Center for Biotechnology Information, National Library of Medicine, National Institutes of Health, Bethesda, MD, USA. [28]Department of Population Health Sciences, Weill Cornell Medicine, New York, NY, USA. [29]Hangzhou Cancer Institute, Hangzhou Cancer Hospital, Hangzhou, China. [30]Department of Radiation Oncology, Hangzhou Cancer Hospital, Hangzhou, China. [31]HudsonAlpha Discovery Institute, Huntsville, AL, USA. [32]Department of Medicine, Division of Infectious Diseases, University of California, San Francisco, CA, USA. [33]The Feil Family Brain and Mind Research Institute, Weill Cornell Medicine, New York, NY, USA. [34]These authors contributed equally: Daniel Butler, Christopher Mozsary, Cem Meydan, Jonathan Foox, Joel Rosiene, Alon Shaiber, David Danko. ✉email: res2025@med.cornell.edu; npt2105@cumc.columbia.edu; har2006@med.cornell.edu; mai9037@med.cornell.edu; chm2042@med.cornell.edu

