## [Peer Review File · Nature Communications]

REVIEWER COMMENTS

Reviewer #3 (Remarks to the Author):

The authors have revised their ambitious and multi-pronged characterization of SARS-CoV-2 infection, including (1) a LAMP assay for rapid detection, (2) viral sequencing and phylogenetic analysis of 735 cases, (3) host transcriptional profiling from the same nasal swabs, and (4) epidemiological analysis of ACEi/ARB use. This work covers a tremendous amount of ground – even more so in this revision.

In this revision, they:

- better contextualized their phylogenetic analysis
- added additional comparators and comorbidity considerations into their epidemiological analysis of ACEi/ARB use
- added spatial transcriptomic profiling of autopsy samples
- added HLA type enrichment of COVID patients compared with those in their cohort who tested negative for SARS-CoV-2

MAJOR CRITIQUES:

- The performance of LAMP compared with RT-qPCR would be better clarified by a direct comparison of LAMP fluorescence vs cycle threshold, either % positive by LAMP for discrete Ct bins, or if possible a plot of relative LAMP fluorescence vs Ct (similar to their Fig S4, but with LAMP fluorescence on the y-axis instead of as a colorscale. Both Fig S3c and Fig S4 make its sensitivity look relatively poor at Ct > 30, and yet they report Se > 95% in Fig 1e.

- The authors comment that the interferon responses in SARS-CoV-2 samples were “significantly higher when compared to SARS-CoV-2 negative samples that harbored other respiratory viruses (Fig 4 a,b)” – but it looks from the labels in Fig 4b that the statistical comparison is to all SARS-CoV-2

negative samples, not just those with alternative respiratory viruses. The biological significance of a difference in interferon response would be quite a bit less if the comparators do not all have viral infections. Distinguishing SARS-CoV-2 specific responses vs general antiviral responses is important, but not sufficiently clear as is.

- The authors have considerably expanded their analysis of ACEi/ARB use to incorporate more comorbidities and other antihypertensive comparators. This portion of the paper feels quite distinct from the remainder, both methodologically and thematically, and probably is best reviewed by a biostatistician, as the detailed analysis of their methods for correction of confounding is critical to assess their conclusions, but is beyond my expertise as a reviewer. The chance for additional confounding with these medications and risk factors for COVID severity remains high; the authors do note this in their discussion.

- HLA enrichment (new in this resubmission), Fig S12: rather than reporting p-values, reporting a statistical test that reflects multiple hypothesis testing (eg false discovery rate) would be preferred for HLA type enrichment; their permutation analysis provides an assessment statistical confidence, but eliminates one allele (HLA-B*35) for which they report a significant p-value. Also, and perhaps more importantly, given disparities in both incidence and severity by race/ethnicity, the biological significance of this HLA enrichment is unclear as it could simply be a proxy for ethnicities that are at greater risk for exposure – this should be commented on.

- Discussion, p11: ACEi/ARBs do not target the ACE2 receptor; the end of the first paragraph on p11 seems to suggest that they do.

MINOR CRITIQUES:

- Fig 4c legend still refers to an intersecting heatmap, but still displays a volcano plot

- Spatial transcriptomics (new in this resubmission), Fig 6c: the authors should clarify these “normal” samples, referred to in-text as “donors”. Are these from autopsies of patients who died of non-pulmonary causes? From living donors undergoing lung biopsy? Other? Neither the text nor the Fig legend (Fig 6 or S11) makes this clear. A comparison to other viral lung infections would be optimal if possible, at least from the literature.

Reviewer #5 (Remarks to the Author):

The authors have generated a very impressive amount of data and analysed it thoroughly. As a result of the review process the authors have revised their article and all the major issues have been responded to. The design model for their transcriptomics analysis and statistics are all appropriate. It is good to see all of the reviewers suggestions/comments have been considered carefully.

Reviewer #6 (Remarks to the Author):

I have been asked to review the statistical analysis of the clinical data. This task is complicated by the poor reporting by the authors.

It is not clear what patients were used for what analysis. The main text reports they have a cohort of $n=50\,821$ patients in two hospitals. The discussion also mentions an analysis of $>50\,000$ patient records. The methods section however describes a cohort of $n=8\,856$ patients with suspected infection of whom $n=4\,829$ were infected. And also data for $n=90\,989$ patients from 2019, presumably before SARS-CoV-2 was present or widespread in New York. However, Table 1 says that $n = 17\,812 + 3\,500 + 23\,578 + 2\,464 = 47\,354$ people were tested for infection, that $n = 4\,697 + 1\,252 + 2\,878 + 479 = 9\,306$ were confirmed to be infected. So: 50k or 90k patients? 9k or 5k infections? 47k or 9k tested?

In my view the paper can't even be reviewed until the reporting is done thoroughly. Who can tell what the results mean if the basics about who was analysed are not set out clearly? The Strobe checklist would be a good place to start this endeavour.

Other issues abound: What were the 2019 patients used for? The main analysis of clinical data has three parts: of risk of being a case, which is clearly 0 for 2019 patients regardless of their profiles, and then of outcomes among cases, again not possible for 2019 patients. So what are they contributing?

In the analysis, the authors say they have scaled quantitative variables to fall on the range 0 to 1 to allow for comparisons. I don't think this is really appropriate: if the distributions differ, the

coefficients are not in fact comparable with such rescaling (instead, standardisation should aim to equalise the standard deviation, eg to set them to have mean 0 and sd 1).

The methods section needs to describe how correlated variables were removed.

Supplementary table 6 to 9 were missing, which reportedly contain some of the results from the clinical data analysis, so I have not reviewed them.

Overall it seems like the focus on the cool transcriptome work led to neglect of the clinical data. I suggest they either remove this part or invest some time in reporting it properly.

In two minor unrelated points: the title is misleading since they have not done a spatial profiling of infection but a spatial transcriptome profiling of infection. Readers might expect to see maps of NY boroughs with the current phrasing. And, paragraph 3 in the intro contains a non-sequitur. "As NY was the epicentre... we did some science". The science doesn't seem to follow from a public health problem in a city.

Response to reviewers for Shotgun Transcriptome, Isothermal, and Spatial Omics Profiling of SARS-CoV-2 Infection (NCOMMS-20-35521A)."

Reviewer #3 (Remarks to the Author):

The authors have revised their ambitious and multi-pronged characterization of SARS-CoV-2 infection, including (1) a LAMP assay for rapid detection, (2) viral sequencing and phylogenetic analysis of 735 cases, (3) host transcriptional profiling from the same nasal swabs, and (4) epidemiological analysis of ACEi/ARB use. This work covers a tremendous amount of ground – even more so in this revision.

We thank the reviewer for their positive comments and review of our revised manuscript.

In this revision, they:

- better contextualized their phylogenetic analysis
- added additional comparators and comorbidity considerations into their epidemiological analysis of ACEi/ARB use
- added spatial transcriptomic profiling of autopsy samples
- added HLA type enrichment of COVID patients compared with those in their cohort who tested negative for SARS-CoV-2

We thank the reviewer for their positive comments and summary.

MAJOR CRITIQUES:

- The performance of LAMP compared with RT-qPCR would be better clarified by a direct comparison of LAMP fluorescence vs cycle threshold, either % positive by LAMP for discrete Ct bins, or if possible a plot of relative LAMP fluorescence vs Ct (similar to their Fig S4, but with LAMP fluorescence on the y-axis instead of as a colorscale. Both Fig S3c and Fig S4 make its sensitivity look relatively poor at Ct > 30, and yet they report Se > 95% in Fig 1e.

We now include a new supplementary figure that shows the LAMP fluorescence vs. Ct values (new Supplemental Figure 4b, \$R^2 = -0.74\$ ). As expected, the higher Ct values correlate with the

lower LAMP output. Of note, the 95% in Fig. 1e was for technical reproducibility from controls, but indeed clinical samples were more variable (Supp Fig 3). Also, we have clarified the text to note that the performance of the assay is indeed a function of viral abundance, now stating:

“Of note, we obtained similar performance on bulk oropharyngeal swab lysate, including increasing reaction sensitivity as a function of viral copy number, but with deteriorating performance at Ct>30.”

We note that, since our first submission, there have been a wealth of papers that now compare LAMP to RT-PCR (most of which that use our exact same primer set), and we have updated our paper with these citations and direct the reader to the literature for additional examples of such comparisons. This includes:

<https://stm.sciencemag.org/content/12/556/eabc7075>

<https://www.medrxiv.org/content/10.1101/2020.06.30.20142935v4>

We also note that our RT-LAMP test has recently been authorized for home collection (in collaboration with Color) and COVID-19 testing:

<https://www.fda.gov/media/138249/download>

- The authors comment that the interferon responses in SARS-CoV-2 samples were “significantly higher when compared to SARS-CoV-2 negative samples that harbored other respiratory viruses (Fig 4 a,b)” – but it looks from the labels in Fig 4b that the statistical comparison is to all SARS-CoV-2 negative samples, not just those with alternative respiratory viruses. The biological significance of a difference in interferon response would be quite a bit less if the comparators do not all have viral infections. Distinguishing SARS-CoV-2 specific responses vs general antiviral responses is important, but not sufficiently clear as is.

We have clarified this distinction in the text, and we now note that the comparison is just those without any known respiratory viruses. To detail virus-specific responses in depth, we think a follow-up paper with larger sample sizes for each viral infection would serve this question better, and we are collecting patients with these other viruses at our hospital, but this will not be ready in time for this revision.

- The authors have considerably expanded their analysis of ACEi/ARB use to incorporate more comorbidities and other antihypertensive comparators. This portion of the paper feels quite distinct from the remainder, both methodologically and thematically, and probably is best

reviewed by a biostatistician, as the detailed analysis of their methods for correction of confounding is critical to assess their conclusions, but is beyond my expertise as a reviewer. The chance for additional confounding with these medications and risk factors for COVID severity remains high; the authors do note this in their discussion.

Please see below for a full set of statistical reviews and our responses and updates for Reviewer #6.

- HLA enrichment (new in this resubmission), Fig S12: rather than reporting p-values, reporting a statistical test that reflects multiple hypothesis testing (eg false discovery rate) would be preferred for HLA type enrichment; their permutation analysis provides an assessment statistical confidence, but eliminates one allele (HLA-B*35) for which they report a significant p-value. Also, and perhaps more importantly, given disparities in both incidence and severity by race/ethnicity, the biological significance of this HLA enrichment is unclear as it could simply be a proxy for ethnicities that are at greater risk for exposure – this should be commented on.

We have clarified our explanation of *HLA* association findings to firmly distinguish first-pass candidate findings (previously reported as raw p-values) from those that withstood withstanding permutation-based correction for false discovery. Indeed, to account for any spurious enrichment from the two-tailed Fisher exact, we used the nonparametric permutation testing to generate empirical p-values for these loci, which is an established method for testing statistical significance. The HLA-B*35 locus only passed significance in one test, which was noted. In parallel, we have clarified our description of permutation methods themselves, in the main text and supplemental figure.

Also, we agree on the need for cautious interpretation, given all the possible confounding variables, and we have updated the testing for the HLA enrichment and added this to the results section:

“We caution that these findings neglect linkage-based interdependence of type frequencies among distinct *HLA* genes; potential ancestry-confounded variability in infection incidence or ascertainment; imprecision of diplotype inference; and any unassayed infection- or transcription-relevant sequence variation distinctive to particular haplotypes within surveyed haplogroups, including in MHC genes other than class I *HLA* loci.”

- Discussion, p11: ACEi/ARBs do not target the ACE2 receptor; the end of the first paragraph on p11 seems to suggest that they do.

We have edited the paragraph to clarify this point. Thank you.

MINOR CRITIQUES:

- Fig 4c legend still refers to an intersecting heatmap, but still displays a volcano plot

We have updated the legend to denote volcano plot.

- Spatial transcriptomics (new in this resubmission), Fig 6c: the authors should clarify these “normal” samples, referred to in-text as “donors”. Are these from autopsies of patients who died of non-pulmonary causes? From living donors undergoing lung biopsy? Other? Neither the text nor the Fig legend (Fig 6 or S11) makes this clear. A comparison to other viral lung infections would be optimal if possible, at least from the literature.

Normal samples in these figures refer to tissues from patients who were lung transplant donors. We have clarified this both in-text and in legends, saying they are from “excess lung material from healthy lung transplant donors.”

We thank the reviewer for the careful reading and helpful suggestions.

Reviewer #5 (Remarks to the Author):

The authors have generated a very impressive amount of data and analysed it thoroughly. As a result of the review process the authors have revised their article and all the major issues have been responded to. The design model for their transcriptomics analysis and statistics are all appropriate. It is good to see all of the reviewers suggestions/comments have been considered carefully.

We thank the reviewer for the positive comments and review of our revised manuscript.

Reviewer #6 (Remarks to the Author):

I have been asked to review the statistical analysis of the clinical data. This task is complicated by the poor reporting by the authors.

It is not clear what patients were used for what analysis. The main text reports they have a cohort of $n=50\,821$ patients in two hospitals. The discussion also mentions an analysis of $>50\,000$ patient records. The methods section however describes a cohort of $n=8\,856$ patients with suspected infection of whom $n=4\,829$ were infected. And also data for $n=90\,989$ patients from 2019, presumably before SARS-CoV-2 was present or widespread in New York. However, Table 1 says that $n = 17\,812 + 3\,500 + 23\,578 + 2\,464 = 47\,354$ people were tested for infection, that $n = 4\,697 + 1\,252 + 2\,878 + 479 = 9\,306$ were confirmed to be infected. So: 50k or 90k patients? 9k or 5k infections? 47k or 9k tested?

- We apologize for our confusing reporting. We have now removed these errors and made our reporting more consistent throughout results, discussion, and methods. Table 1 and the text should now both be much more clear to a reader. Specifically:
 - Two separate cohorts were used for the transcriptomic and clinical analyses. The transcriptomic analysis used 669 individuals, while the clinical analysis used 50,821 individuals (23,170 from Columbia and 27,651 from Cornell).
 - After reflecting on the interpretation and the possible issues raised by the reviewer, we decided to remove the comparison with a 2019 clinical cohort (see also below).

In my view the paper can't even be reviewed until the reporting is done thoroughly. Who can tell what the results mean if the basics about who was analyzed are not set out clearly? The Strobe checklist would be a good place to start this endeavour.

We agree with the reviewer. We have completed Strobe checklist with our resubmission.

Other issues abound: What were the 2019 patients used for? The main analysis of clinical data has three parts: of risk of being a case, which is clearly 0 for 2019 patients regardless of their profiles, and then of outcomes among cases, again not possible for 2019 patients. So what are they contributing?

We appreciate the reviewer's comment. We had been using these patients as a reference for patients without any recorded test. Upon reflection, however, these comparisons are unnecessary and difficult to interpret. They have been removed to help clarify our methods and the paper overall, to focus just on the COVID-19 cohort and timeframe.

In the analysis, the authors say they have scaled quantitative variables to fall on the range 0 to 1 to allow for comparisons. I don't think this is really appropriate: if the distributions differ, the

coefficients are not in fact comparable with such rescaling (instead, standardization should aim to equalize the standard deviation, eg to set them to have mean 0 and sd 1).

- We agree with the suggestion that zero mean unit variance scaling is more useful for interpreting the coefficients of quantitative variables. However, instead of modifying the analysis, we removed estimates of covariate effect from our tables. Four reasons motivated this decision.
 - First, while the analysis in our initial submission included various quantitative variables, age is the only quantitative variable in our current analysis, and age was not scaled. This confusion is due to our including additional paragraphs from the previous revision.
 - Second, since covariate associations are not adjusted for confounding the same way as the variable of interest, these are not estimates which can be interpreted independently. Estimates for all covariates, for example age, are conditional on drug exposure and relevant confounders for drug exposure, not the relevant confounders for age.
 - Third, the method of scaling employed does not affect the estimates of drug effects, our actual estimates of interest.
 - Fourth, as our analysis was run separately at different institutions using separate electronic health record systems, restricted computational frameworks, and numerous compatibility challenges, a minor modification to the analysis entails a prohibitive time cost.

The methods section needs to describe how correlated variables were removed.

The final analysis we report did not include a removal of correlated variables. This language was part of an earlier revision and methods section and was unfortunately retained in the final version. We have removed this language and updated the section.

Supplementary table 6 to 9 were missing, which reportedly contain some of the results from the clinical data analysis, so I have not reviewed them.

These tables were uploaded into the online system, but they did not transit correctly. To simplify this, we have condensed these tables into one (Supp Table 6) and included it in the revision, as well as a new clinical analysis methods section (Supplemental Methods).

Overall it seems like the focus on the cool transcriptome work led to neglect of the clinical data. I suggest they either remove this part or invest some time in reporting it properly.

We hope these latest revisions help to clarify the reporting and our statistical analyses of the clinical data. We believe that linking the molecular data to the clinical data for this work provides a manuscript of greater depth and thoroughness.

In two minor unrelated points: the title is misleading since they have not done a spatial profiling of infection but a spatial transcriptome profiling of infection. Readers might expect to see maps of NY boroughs with the current phrasing. And, paragraph 3 in the intro contains a non-sequitur. "As NY was the epicentre... we did some science". The science doesn't seem to follow from a public health problem in a city.

We have updated the title to clarify that it was spatial omics (transcripts and protein) profiling, now stated as "**Shotgun Transcriptome, Isothermal, and Spatial Omics Profiling of SARS-CoV-2 Infection.**"

For the introduction, we agree. We have changed the introductory clause for that 3rd paragraph to now lead with:

"To better understand the impact and progression of SARS-CoV-2 infection, we applied a multi-platform and molecular diagnostic approach to samples collected during the outbreak in NYC."

REVIEWERS' COMMENTS

Reviewer #3 (Remarks to the Author):

“Shotgun transcriptome, isothermal, and spatial omics profiling of SARS-CoV-2 infection”

Butler, Mozsary, Meydan, Foox, Rosiene, Shaiber, Danko, et al (Schwartz, Tatonetti, Rennert, Imileinski, Mason).

Revised and resubmitted to Nature Communications, Nov 2020

The authors have once again revised their multi-pronged characterization of SARS-CoV-2 infection, clarifying several points raised by reviewers and adding appropriate caveats. For the non-biostatistical (i.e., non-ACEi/ARB related) portions of the manuscript, they have addressed critiques adequately and strengthened the manuscript as a result. Analysis of the ACEi/ARB correlations remains outside of this reviewer’s expertise, though even in the absence of this section, the paper could stand on its merits; this portion is not necessary for the omics portions of the paper and in some ways still feels as though it could be built upon and clarified in a separate manuscript. Even with the authors caveats, since their conclusions about ACEi/ARB point to a possible use case, they must be made with extreme care.

REMAINING SUGGESTIONS:

- In their response to reviewers, the authors state that “Of note, the 95% in Fig. 1e was for technical reproducibility from controls, but indeed clinical samples were more variable (Supp Fig 3).” Yet in both text (lines 165-167) and figure legends (Fig 1e, Supp Fig 2), results are touted as “overall sensitivity of 95.6%” (line 166) from “201 patients” (Fig 1e legend). If this report of 95% sensitivity is indeed for technical reproducibility and NOT for clinical performance, then this must be made more clear in both text and figure legends.
- For Supp Fig 4, the addition of panel b is helpful. Is it possible to add a threshold line on the y axis for LAMP, similar to the one on the x-axis for RT-qPCR? Also, the stacked-points make the specificity look worse than it probably is by emphasizing LAMP+, qPCR- samples whereas the concordant samples all presumably stack at low LAMP values – perhaps a distribution of LAMP values for RT-qPCR positive and negative samples (eg at right) would help clarify?
- In their response to reviewers, the authors state that “We have clarified this distinction in the text, and we now note that the comparison is just those without any known respiratory viruses”. Yet on lines 302-305, when Fig 4a,b is cited, they still state the opposite: “a common interferon response across all ranges of viral levels, which was significantly higher when compared to SARS-CoV-2

negative samples that harbored other respiratory viruses". This is what Fig 4b shows – the only stat sig comparison shown was to those without respiratory viruses detected (which is unsurprising – but still worth reporting). In Fig 4a, the "other respiratory viruses" subclass (pink) appears roughly similar in magnitude of response to the "low" SARS-CoV-2 subclass (yellow) – perhaps also unsurprising. The text on lines 302-305 should be clarified as they indicate in their response to reviewers. The finer distinctions between COVID-19 and other respiratory viral illnesses does appear to be ripe for further study in a subsequent paper, as the authors indicate.

- If ACEi/ARB findings pass statistical review, the Discussion should be rewritten with an even stronger caveat: prospective, randomized clinical trials should certainly be performed before any uniform use of these medications could be considered (not "may be necessary", as the authors state on line 476).

Reviewer #6 (Remarks to the Author):

Thank you for addressing my comments. The description of the clinical cohort is much clearer now.

I have been asked to comment further on the last point raised by Reviewer #3. I agree with the other reviewer that the authors should be cautious in their phrasing. The results and discussion do tend to impart more certainty of a causal relationship than is warranted. To hasten the process, I have suggested in the provided attachment revisions to the offending paragraphs that I think overcome any concerns about over-selling the work.

I also agree with the reviewer that indeed the authors could drop the clinical cohort and it would still be a very substantial contribution to the literature. I do not believe it necessary to do so however.

We would like to thank Reviewer #6 for the suggested modifications outlined below. We have updated the manuscript to incorporate all these suggestions.

Suggested modifications given by Reviewer #6:

RESULTS

To address this, we analyzed ACEI/ARB use and severe COVID-19 outcomes in an observational cohort of individuals (n=50,821) suspected of SARS-CoV-2 infection at New York Presbyterian Hospitals, comprising 23,170 patients from Columbia University Irving Medical Center (NYP-CUIMC) and 27,651 patients from Weill Cornell Medical Center (NYP-WCMC; Table 1). At both sites, we found evidence that ACEI/ARB use was associated with lower rates of intubation (CUIMC Hazard ratio, HR=0.79, 95% confidence interval, CI: 0.68-0.91, WCMC HR=0.62, CI: 0.48-0.81) and lower risk of death (CUIMC HR=0.66, CI: 0.58-0.75, WCMC HR=0.73, CI: 0.56-0.95) following confirmed SARS-CoV-2 infection (Figure 5). Each comparison used propensity matching (PSM) and Cox proportional hazards models with covariate adjustments for age, sex, race, ethnicity, drug indications, and relevant comorbidities (see methods; Table 1).

ACEI/ARB have a number of indications in addition to hypertension, so confounding remains a challenge. For comparison, we repeated the analysis in a cohort of individuals with recent exposure to one of four antihypertensive medication classes: ACEI/ARB, beta blockers (BB), calcium channel blockers (CCB), and thiazide/thiazide-like diuretics (THZ)). Evaluating each drug class separately using PSM and covariate adjustment, we found the rate of intubation was consistently lower for ACEI/ARB exposure (CUIMC HR=0.74, CI: 0.64-0.85, WCMC HR=0.73, CI: 0.56-0.95) and higher for THZ exposure (CUIMC HR=1.96, CI: 1.67-2.31, WCMC HR=1.90, 1.42-2.55). Conversely, we found no statistically significant associations for CCB (CUIMC HR=0.92, 0.81-1.05, WCMC HR=1.04, CI: 0.83-1.30), and only weak evidence of positive associations between intubation and BB exposure (CUIMC HR=1.16, CI: 1.03-1.31, WCMC HR=1.04, CI: 0.84-1.27). Meanwhile, we found consistently lower rates of death among patients with ACEI/ARB (CUIMC HR=0.82, CI: 0.71-0.93, WCMC HR=0.67, CI: 0.52-0.87), BB (CUIMC HR=0.81, CI: 0.71-0.91, WCMC HR=0.74, CI: 0.61-0.91), and CCB (CUIMC HR=0.63, CI: 0.55-0.72, WCMC HR=0.69, CI: 0.55-0.86) exposures, while THZ exposure had harmful associations (CUIMC HR=1.03, CI: 0.87-1.22, WCMC HR=1.43, CI: 1.04-1.96). In summary, among SARS-CoV-2-infected individuals with recent exposure to antihypertensive drugs, ACEI/ARBs were the only class which were associated with lower risk of intubation, while ACEI/ARB, BB, and CCB use was negatively associated with death. (Complete estimates are provided in Supplemental Table 6.)

DISCUSSION

We examined the risk of medication use and comorbidities in the context of ACEI or ARB. Our retrospective clinical analysis of 50,821 patient records shows that patients with ACEI/ARB exposure had lower risk of severe COVID-19 outcomes, after adjusting for demographics, indications, and other relevant comorbidities. A recent study reported a protective effect of ACEI/ARB exposure on mortality (Zhang et al., 2020). The associations we found support this finding, and we found similar associations for mechanical respiration requirements among infected patients. Further, our data did not show that other antihypertensive drug classes had any association with mechanical respiration requirements, though BB and CCB use was associated with lower risk of death as well. While our data suggest a protective association between ACEI/ARB use and severe COVID-19 from two

large New York hospitals (Columbia and Weill Cornell), our results are still preliminary, and caution should be taken **in interpreting them. Because of the risk of residual confounding, despite our efforts to adjust for important confounders,** prospective clinical trials **would** be necessary **before clinical guidelines should be changed.** For example, if some patients are more susceptible because they already express high levels of ACE2, this could help with targeting the ACE pathway in these patients as a prophylactic method. However, if the cells respond to infection with ACE2 expression, and this leads to the cytokine storm seen in patients, then this could be used as a downstream treatment (post-infection), for when ACE2 interacts with TMPRSS2, such as the ongoing trials with camostat mesylate (Hoffman et al., 2020).